# TGA: True-to-Geometry Avatar Dynamic Reconstruction

**Bo Guo**[1]     **Sijia Wen**[1]*     **Ziwei Wang**[1]     **Yifan Zhao**[2]

[1]Beijing Advanced Innovation Center for Future Blockchain and Privacy Computing,
School of Artificial Intelligence, Beihang University
[2]State Key Laboratory of Virtual Reality Technology and Systems,
SCSE&QRI, Beihang University
`{keaibb, sijiawen, wangziwei26, zhaoyf}@buaa.edu.cn`

## Abstract

Recent advances in 3D Gaussian Splatting (3DGS) have improved the visual fidelity of dynamic avatar reconstruction. However, existing methods often overlook the inherent chromatic similarity of human skin tones, leading to poor capture of intricate facial geometry under subtle appearance changes. This is caused by the affine approximation of Gaussian projection, which fails to be perspective-aware to depth-induced shear effects. To this end, we propose True-to-Geometry Avatar Dynamic Reconstruction (TGA), a perspective-aware 4D Gaussian avatar framework that sensitively captures fine-grained facial variations for accurate 3D geometry reconstruction. Specifically, to enable color-sensitive and geometry-consistent Gaussian representations under dynamic conditions, we introduce the Perspective-Aware Gaussian Transformation that jointly models temporal deformations and spatial projection by integrating Jacobian-guided adaptive deformation into the homogeneous formulation. Furthermore, we develop Incremental BVH Tree Pivoting to enable fast frame-by-frame mesh extraction for 4D Gaussian representations. A dynamic Gaussian Bounding Volume Hierarchy (BVH) tree is used to model the topological relationships among points, where active ones are filtered out by BVH pivoting and subsequently re-triangulated for surface reconstruction. Extensive experiments demonstrate that TGA achieves superior geometric accuracy. Project page: `https://superkeaibb.github.io/TGA/`.

## 1   Introduction

The demand for personalized, high-fidelity reconstruction of human faces and heads under complex facial movements is fundamental to a wide range of applications, including digital twins, film production, graphics simulation, and entertainment. In particular, acquiring dynamic and geometrically accurate 3D head reconstructions from multi-view recordings is a common requirement for generating digital and virtual replicas of real individuals.

Despite recent advances in high-fidelity 3D Gaussian Splatting (3DGS) [1]-based avatar reconstruction, existing methods [2, 3, 4] often overlook the inherent chromatic similarity of human skin tones. Specifically, under varying viewpoints and frame transitions, subtle and gradual facial expression changes remain challenging for vanilla 3DGS. The root cause lies in the limited *perspective-awareness* (we will discuss in Sec. 3.2) of affine approximation used in Gaussian projection, which compromises accurate color blending. As a result, the simplistic projection model struggles to maintain chromatic consistency and further geometric fidelity for dynamic face modeling.

---

*Corresponding author

39th Conference on Neural Information Processing Systems (NeurIPS 2025).

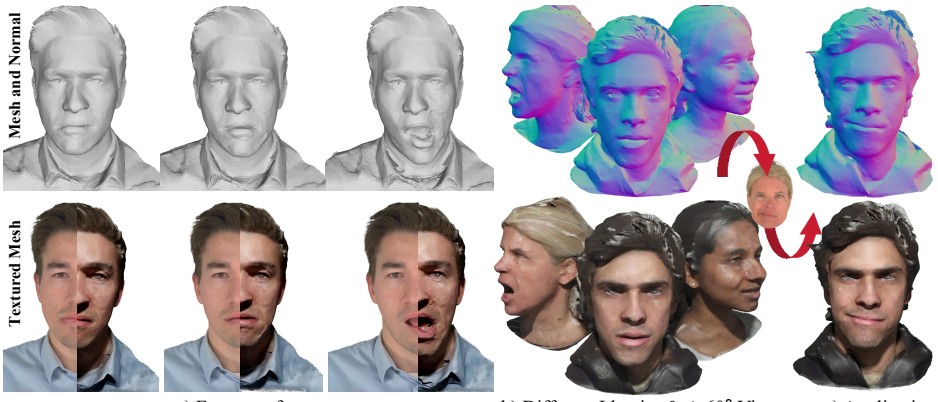

Mesh and Normal

Textured Mesh

a) Frame-to-frame      b) Different Identity & ± 60° Views      c) Application

Figure 1: Example results of our TGA. Our method generates (a) high-fidelity, frame-by-frame textured meshes from multi-view videos, (b) captures avatar-specific details across wide viewing angles, and (c) delivers realistic cross-reenactment performance.

In response, we propose True-to-Geometry Avatar Dynamic Reconstruction (TGA), a *perspective-aware* 4D Gaussian avatar framework that sensitively captures fine-grained facial appearance variations to enable geometrically accurate 3D mesh reconstruction from 4D Gaussian representations.

We first introduce a Perspective-Aware Gaussian Transformation that jointly models temporal deformations and spatial projection effects, enhancing *perspective-awareness* to subtle changes in avatar facial appearance. Traditional 3DMM [5, 6]-based Gaussian methods [2, 3, 4, 7] warp primitives based on the area of their parent triangle across time steps. Such uniform scaling leads to either extending or shrinking Gaussian coverage, resulting in over- or under-blending of colors in facial regions. To address this, we apply Jacobian-guided deformation to adaptively warp each Gaussian according to the directional variation of its parent triangle, ensuring precise color coverage across dynamic frames. Furthermore, to be *perspective-aware* to the projection process for better capture of the intricate geometry under subtle changes in avatar skin tone and facial expressions, we adopt a homogeneous formulation in place of the insufficient affine approximation in vanilla 3DGS, enabling perspective-consistent projections. The Jacobian-guided deformation is jointly incorporated into the homogeneous formulation. As a result, color blending becomes more chromaticity-sensitive, reinforcing geometry-color alignment and providing a reliable foundation for downstream mesh extraction.

Based on the obtained Gaussian avatar field, we introduce an opacity field [8] for surface extraction. The geometrically accurate avatar surface is extracted by directly identifying a level set of an opacity-guided signed distance field on tetrahedra, which are triangulated from Gaussians and their bounding points. To enable fast dynamic mesh extraction, we design a straightforward yet effective Incremental BVH Tree Pivoting approach to adaptively update the tetrahedral grids. Specifically, we dynamically organize the bounding volume hierarchy (BVH) of Gaussians in a binary tree, which models the topological relationships among primitives. As the BVH tree updates by branch rotation according to Gaussian dynamics, the BVH is pivoted to simulate the topological movements of Gaussians. Active 3D points—referred to as hopping Gaussians—that contribute to facial expression changes, are filtered out by BVH Pivoting. These filtered regions are then incrementally triangulated, thereby accelerating the surface extraction process.

Overall, our contributions are the following:

- We propose a *perspective-aware* 4D Gaussian avatar framework that captures intricate geometry under subtle facial variations, by integrating Jacobian-guided adaptive deformation with homogeneous projection to enable Perspective-Aware Gaussian Transformation.

- We design an Incremental BVH Tree Pivoting approach, which filters out and re-triangulates hopping Gaussians. This enables fast and adaptive surface extraction by focusing computation on active regions undergoing facial expression changes.

- We empirically demonstrate the advanced performance of the proposed method, demonstrating significant improvements in reconstruction accuracy, dynamic capability, training efficiency, and inference time.

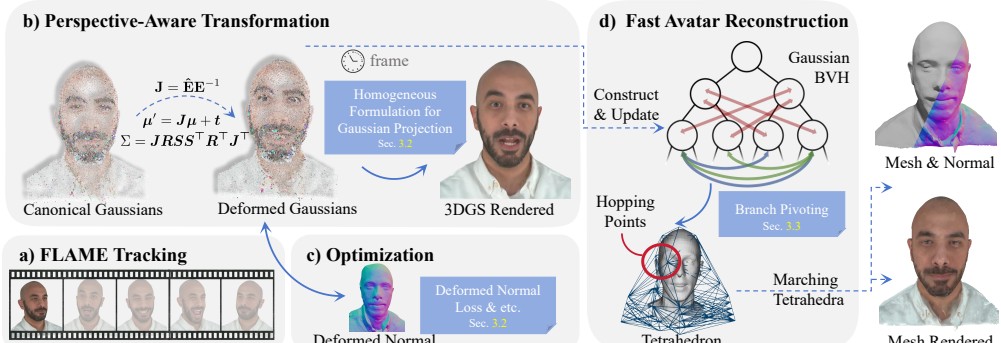

Figure 2: **Method Overview**: Given multi-view RGB sequences, we first track facial dynamics with FLAME (a). During the Perspective-aware Transformation (b), we apply Jacobian-guided deformation and homogeneous projection for accurate geometric modeling. After optimization (c), we build and dynamically update a Gaussian BVH (d), where BVH pivoting adaptively filters hopping points, enabling geo-accurate surface extraction via Marching Tetrahedra (rightmost column).

## 2 Related Work

**3D Morphable Face Models.** Parametric template models based on PCA have become fundamental in computer graphics and vision for representing human body geometry, including the face [6, 9, 10] and head [5, 11], with extensions to the neck [12] and the full body [13]. To overcome the rigid linearity of PCA, more recent approaches [14, 15, 16, 17, 18, 19, 20, 21] replace traditional PCA-basis underlying classical mesh-based 3DMMs. Furthermore, neural-based methods [22, 23, 24] enhance expression realism with continuous, implicit morphable representations of geometry.

**Dynamic Avatar Representations.** Avatars have inherent dynamics, especially when performing actions such as smiling or speaking, which are accompanied by significant topological changes. This makes the representation of dynamic scenes more complex and challenging. Neural Radiance Fields (NeRFs) [25, 26, 27]-based methods [25, 28, 29, 30, 31, 32, 33] can capture temporal changes and model such dynamics but not computationally feasible. 3DGS [1]-based dynamic methods [34, 35, 36, 37, 38, 39, 40] have emerged as a more efficient alternative, but remain insensitive to chromatic variations due to their affine approximation. Building upon the 3DGS paradigm, we enhance it with a Perspective-Aware Gaussian Transformation module for improved dynamic modeling.

**Human Head Reconstruction.** Previous works [41, 42, 43, 44, 45] have explored NeRF-based volume rendering to model avatar heads with detailed appearance. Recent approaches [46, 47, 48, 3, 49, 40, 50] incorporate implicit deformation fields to capture frame-wise Gaussian motion. Meanwhile, another line of work [51, 52, 2, 53, 4] explicitly rig Gaussians to 3DMM-based meshes for controllable facial animation. Topo4D [54] further reconstructs dynamic meshes and high-fidelity textures via topology-bound Gaussians, NPGA [55] leverages neural parametric head models [56] for learned forward deformations, ScaffoldAvatar [57] employs patch-based expressions with hierarchical Gaussian splatting for high-fidelity avatars. While most 3DGS-based methods target photorealistic rendering and animation, Topo4D [54], SurFhead [4], and our work focus on geometry-accurate facial mesh reconstruction. Specifically, our method builds upon the explicit 3DMM–3DGS binding and deformation framework for better controllability.

## 3 Method

With the goal of achieving geometrically accurate 4D avatar reconstruction in our mind, we first focus on capturing subtle variations in facial appearance while ensuring deformation consistency across frames (Sec. 3.2). By leveraging trained Gaussians with high sensitivity to facial features, our approach further enables fast and precise avatar mesh extraction (Sec. 3.3). We start with the overview of GaussianAvatars [2] and also define the main symbols(Sec. 3.1).

### 3.1 Preliminary

**Representation.** GaussianAvatars [2] associates each triangle of the FLAME mesh [5], tracked by VHAP [58], with a 3D Gaussian and moves coherently with its corresponding parent-triangle

across time steps. Specifically, each Gaussian is parameterized by a center position $\mu$, a positive-definite, diagonal matrix scaling matrix $S$, and a rotation matrix $R$ on the neutral FLAME mesh [5]: $G(\mathbf{x}) = e^{-\frac{1}{2}(\mathbf{x}-\mu)^\top \Sigma^{-1}(\mathbf{x}-\mu)}$ where $\Sigma = RS^2R^\top$. Besides, the Gaussian primitive has appearance properties, a prior opacity $\alpha$ and color $\mathbf{c}$. To deform the canonical 3D Gaussian to the posed space, the vanilla 3DMM-Gaussians method transforms its position and covariance attributes according to its parent:

$$\mathbf{R}' = \mathbf{rR} \quad \mu' = k\mathbf{r}\mu + \mathbf{t} \quad \mathbf{S}' = k\mathbf{S}, \tag{1}$$

where the isotropic scalar k is derived from the triangle's relative extent, $t$ and $r$ are the barycenter and relative rotation matrix of the parent-triangle, respectively.

**Rendering.** Given the 3D representation $\theta = \{\mu, \Sigma, \alpha, \mathbf{c}\}$, the trainable parameters are optimized through the following differentiable rendering function

$$\tilde{\mathbf{C}}(\mathbf{p}) = \sum_{n=1}^{N} c_n \alpha_n \tilde{D}(\mu, \Sigma, \gamma) \prod_{m=1}^{n-1} (1 - \alpha_m \tilde{D}(\mu, \Sigma, \gamma)), \tag{2}$$

where $\tilde{\mathbf{C}}(\mathbf{p})$ is the rendering color at pixel $\mathbf{p}$ of rendered image $\tilde{\mathbf{C}}$, and $\tilde{D}(\mu, \Sigma, \gamma)$ is a divergence of view ray $\gamma$ from $\theta$ computed from the projected 2D Gaussians by EWA volume splatting [59].

### 3.2 Perspective-Aware Gaussian Transformation

Our goal is to capture intricate geometric details under subtle variations in avatar skin tone and facial expression. Previous 3DGS-based methods [2, 3, 4, 7] rely solely on affine approximations when projecting Gaussians onto 2D image planes. However, as illustrated by the blue circle in Fig. 3, this naive projection leads to unreliable alpha blending, resulting in chromatic inconsistencies and spatial distortions. To address this issue, it is essential to revisit the projection mechanism from the perspective of *Perspective Awareness*.

As illustrated by the affine-approximated projected blue Gaussian in Fig. 3, even though its center is correctly splatted using a perspective projection (as shown in 1), the outermost isocontour remains significantly misaligned. This misalignment arises because the affine projection naively projects the Gaussian orthogonally onto the image plane, neglecting the depth information in the Gaussian covariance. Such a simplification fails to be perspective-aware enough to subtle chromatic variations. To achieve true *perspective awareness*, it is imperative to 'look into' the Gaussian itself and fully exploit its depth dimension—only then can we faithfully simulate perspective projection.

Specifically, we first introduce a Jacobian-based deformation mechanism to adaptively warp Gaussians across frames. This deformation is integrated into a homogeneous formulation that preserves depth information, enabling a *perspective-aware* projection. As a result, the model can sensitively capture intricate geometric details driven by subtle chromatic variations in facial expressions.

**Jacobian Gradient for Adaptive Deformation.** While the rigging method in Eq. 1 is computationally efficient, it struggles to preserve the chromatic consistency across frames. Specifically, its rigid and linear isotropic scaling uniformly extends or shrinks Gaussians according to their parent-triangle's property (Fig. 3 (a)), leading to under-blending discontinuity across frames. To address this, inspired by [60], we introduce an advanced warping method for Gaussians:

$$\mathbf{JE} = \hat{\mathbf{E}} \quad \mathbf{J} = \hat{\mathbf{E}}\mathbf{E}^{-1}, \tag{3}$$

where $\mathbf{E}$ is composed of edge direction vectors of the binding triangle, and $\hat{\mathbf{E}}$ is deformed by Eq. 1. Then, Gaussians are warped as follows:

$$\Sigma = \mathbf{JRSS}^\top \mathbf{R}^\top \mathbf{J}^\top \quad \mu' = \mathbf{J}\mu + \mathbf{t}. \tag{4}$$

By doing so, Gaussians gain adaptive and anisotropic deformability (Fig. 3 (b)) and maintain chromatic consistency across frames, making them ready to capture subtle variations.

**Homogeneous Formulation for Gaussian Projection.** To be *perspective-aware* for Gaussian depth dimension and further capture subtle changes in avatar skin tone and facial expressions, we extend prior methods [61, 62] originally designed for 2D Gaussians by introducing a homogeneous formulation that replaces the affine approximation for 3D Gaussians. This accurately models the outermost isocontour under perspective projection and further enables high blending sensitivity to

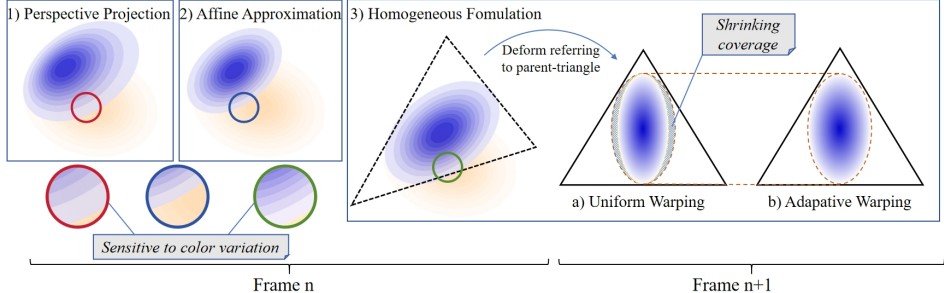

Figure 3: **Perspective-Aware Gaussian Transformation.** Compared with 2) Affine Approximation (fails to capture shear effects and causes inaccurate color blending), our homogeneous formulation compensates for it, while maintaining a sensitive response to subtle color variations like 1) Perspective Projection. For temporal deformation, uniform warping may shrink coverage, but our adaptive warping maintains an accurate spatial footprint.

fine-grained facial appearance changes (green circle in Fig. 3.(2)). The homogeneous transformation for 3D Gaussian to be normalized in a local tangent plane is:

$$\mathbf{H} = \begin{bmatrix} \mathbf{s}_u \mathbf{r}_u & \mathbf{s}_v \mathbf{r}_v & \mathbf{s}_w \mathbf{r}_w & \boldsymbol{\mu} \\ 0 & 0 & 0 & 1 \end{bmatrix} = \begin{bmatrix} \mathbf{RS} & \boldsymbol{\mu} \\ \mathbf{0} & 1 \end{bmatrix}. \tag{5}$$

To jointly account for non-uniform deformations driven by facial expressions across frames, we further refine the Gaussian's in the local metric through a Jacobian-based transformation. This enables adaptive Gaussian deformation while preserving global perspective accuracy. Specifically, we parameterize a viewing ray $\boldsymbol{\gamma}$ passing through a pixel at $(x, y)$ as the intersection of two perpendicular planes: the x-plane $\mathbf{h}_x = (-1, 0, 0, x)^\top$ and the y-plane $\mathbf{h}_y = (0, -1, 0, y)^\top$. The ray is then transformed into the deformed Gaussian's local space by:

$$\mathbf{h}_{u/v} = \left( (\mathbf{MHJ})^{-1} \right)^{-\top} \mathbf{h}_{x/y} = (\mathbf{MHJ})^\top \mathbf{h}_{x/y}, \tag{6}$$

where $\mathbf{M}$ is the transformation matrix from world to screen space. It should be noticed that as combining transformation $\mathbf{MHJ}$, the viewpoint, 3D anisotropic structure of Gaussian and facial expressions, are encoded into perspective projection. After projection, we normalize the homogeneous coordinates to recover 3D positions of the viewing ray in the deformed tangent plane:

$$\mathbf{h}'_u = \left( \frac{\mathbf{h}_u^1}{\mathbf{h}_u^4}, \frac{\mathbf{h}_u^2}{\mathbf{h}_u^4}, \frac{\mathbf{h}_u^3}{\mathbf{h}_u^4}, 1 \right) \quad \mathbf{h}'_v = \left( \frac{\mathbf{h}_v^1}{\mathbf{h}_v^4}, \frac{\mathbf{h}_v^2}{\mathbf{h}_v^4}, \frac{\mathbf{h}_v^3}{\mathbf{h}_v^4}, 1 \right) \tag{7}$$

where $\mathbf{h}'^i_u$ denotes for the i-th component. This step performs a **perspective-divide** by normalizing the homogeneous coordinates (dividing by $\mathbf{h}^4$), which reconstructs the true 3D intersection point between the viewing ray and the deformed tangent plane of the Gaussian in perspective space. Unlike the affine approximation that assumes an orthogonal footprint, this **perspective-divide** "looks into" the Gaussian covariance. Finally, we evaluate the divergence of a camera ray from Gaussian in a straightforward way. Specifically, with

$$\mathbf{m} = (\mathbf{h}'^1_u, \mathbf{h}'^2_u, \mathbf{h}'^3_u) \times (\mathbf{h}'^1_v, \mathbf{h}'^2_v, \mathbf{h}'^3_v) \quad \mathbf{l} = (\mathbf{h}'^1_u, \mathbf{h}'^2_u, \mathbf{h}'^3_u) - (\mathbf{h}'^1_v, \mathbf{h}'^2_v, \mathbf{h}'^3_v), \tag{8}$$

we define the divergence $\mathbf{D}$ derived from the perpendicular distance $\phi^*$ between the Gaussian center and its closest point on the viewing ray in local tangent plane, $\|\mathbf{v}\|$ stand for magnitude of a vector $\mathbf{v}$:

$$\mathbf{D}(\boldsymbol{\mu}, \Sigma, \boldsymbol{\gamma}) = e^{-\frac{1}{2}(\phi^*)^2} \quad \phi^* = \frac{\|\mathbf{m}\|}{\|\mathbf{l}\|}. \tag{9}$$

**Volume rendering.** Then, we obtain screen points by $\mathbf{H}' = \mathbf{MHJ}$, the center $\mathbf{o}$ of projected splat and the outermost bounds $\mathbf{e}$ is computed as:

$$\mathbf{o}_i = \langle \mathbf{f}, \mathbf{H}'_i \cdot \mathbf{H}'_4 \rangle \quad \mathbf{e}_i = \sqrt{\mathbf{o}_i^2 - \langle \mathbf{f}, \mathbf{H}'_i \cdot \mathbf{H}'_i \rangle} \quad \mathbf{f} = \frac{(1,1,1,-1)}{\langle (1,1,1,-1), (\mathbf{H}'_4 \cdot \mathbf{H}'_4) \rangle} \tag{10}$$

where $\mathbf{H}'_i$ is the i-th row of $\mathbf{H}'$, $\langle x, y \rangle$ stands for dot product. The volumetric alpha blending is conducted to compute the rendered color in a *perspective-aware* way:

$$\mathbf{C}(\mathbf{p}) = \sum_{k=1}^{K} \mathbf{c}_k \alpha_k \mathbf{D}(\boldsymbol{\mu}_k, \Sigma_k, \boldsymbol{\gamma}) \prod_{j=1}^{k-1} (1 - \alpha_j \mathbf{D}(\boldsymbol{\mu}_j, \Sigma_j, \boldsymbol{\gamma}))). \tag{11}$$

**Optimization.** We introduce a deformed normal regularization to maintain the learned structure of often-occluded regions such as teeth and eyeballs:

$$\mathcal{L}_{\mathrm{nr}} = \|\mathbf{n^d}_w - \mathbf{n}_w\|_2 \tag{12}$$

where $\mathbf{n}_w$ denotes the Gaussian normal in the $w$-th frame, computed as the inverse camera ray direction $-\boldsymbol{\gamma}$ in the deformed tangent plane and then transformed back to world space: $\mathbf{n}_w = \mathbf{Rot}_w \cdot (-\boldsymbol{\gamma})$, $\mathbf{Rot}_w$ is the rotation component of matrix $\mathbf{MHJ}$. The deformed normal is $\mathbf{n^d}_w = \mathbf{J}_w^{-\top}\mathbf{n}_0$, where $\mathbf{J}_w$ is from Eq. 3 in the $w$-th frame and $\mathbf{n}_0$ is the canonical normal.

Finally, we optimize our model with the following loss:

$$\mathcal{L} = \mathcal{L}_c + \lambda_d\mathcal{L}_d + \lambda_n\mathcal{L}_n + \lambda_s\mathcal{L}_{\mathrm{scaling}} + \lambda_p\mathcal{L}_{\mathrm{position}} + \lambda_{nr}\mathcal{L}_{nr} \tag{13}$$

where $\mathcal{L}_c$ is a combination of photometric loss $\mathcal{L}_{\mathrm{rgb}}$ and a D-SSIM term following 3DGS [1]. To ensure accurate geometry reconstruction, we incorporate geometric loss terms from [8], including depth-distortion $\mathcal{L}_d$ and normal consistency $\mathcal{L}_n$. Toward a better alignment between Gaussians and parent triangles, we use regularization terms $\mathcal{L}_{\mathrm{scaling}}$ and $\mathcal{L}_{\mathrm{position}}$ from [2]. We set the hyperparameters following these works and $\lambda_{nr}$ as 0.01.

### 3.3 Incremental BVH Tree Pivoting for Fast Avatar Reconstruction

To enable rapid and geometry-accurate extraction of avatar meshes from the opacity fields introduced in Sec. 3.2, we adopt an incremental triangulation strategy for dynamic points induced by facial expressions, referred to as 'hopping points'. The core of our approach lies in organizing Gaussians into a dynamic BVH tree, which accurately simulates vertex topological movements through Gaussian BVH pivoting, thereby facilitating efficient and adaptive mesh reconstruction.

#### 3.3.1 Incremental Triangulation

For the initial frame, we employ 3D Delaunay triangulation to generate tetrahedral grids for each Gaussian and its bounding points. We then perform opacity evaluation, defined as the minimum opacity across all visible and relevant training views (depending on different areas in the FLAME model, e.g., the side view for hair) that observe the point. Finally, binary search is conducted over the opacity-SDF field of all tetrahedral grids to locate the zero level set.

For subsequent frames, we employ incremental triangulation guided by hopping points filtering via dynamic Gaussian BVH tree pivoting, as illustrated in Fig. 4, significantly reducing computation time compared to full re-triangulation. Since Gaussian movements fluctuate between frames, applying a fixed threshold to screen Gaussian centers and covariances can result in: 1) incomplete culling, causing a significant degradation in mesh extraction quality; or 2) excessive computation, which reduces the frame rate. Leveraging on our dynamic Gaussian BVH, which accurately simulates the topological movements of vertices, hopping points are swiftly filtered through branch rotation.

#### 3.3.2 Hopping Gaussians Filtering via Dynamic BVH Pivoting

We begin by constructing a static binary radix BVH tree [63] from a given set of 3D Gaussians, where each leaf node represents the tight bounding box of a Gaussian cluster, and each internal node denotes the bounding box encompassing its two child nodes. To dynamically simulate the topological movements of Gaussians over time, we perform pivoting of the Gaussian BVH tree at each frame.

**BVH Pivoting.** To better illustrate this topological simulation, we demonstrate our BVH structure shown in Fig. 4, where yellow "a", red "b", blue "c", and green "d" bounding boxes correspond to facial features, alongside its tree representation of the avatar. As facial expressions occur (e.g., eye opening), these bounding volumes are locally refitted according to the movement of enclosed Gaussians. As the extent of box "d" significantly expands to tightly enclose

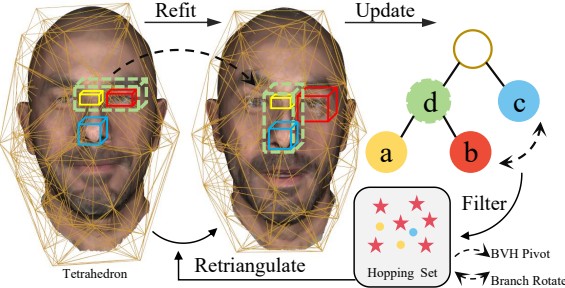

Figure 4: **BVH Pivoting for Hopping Points Filtering.**

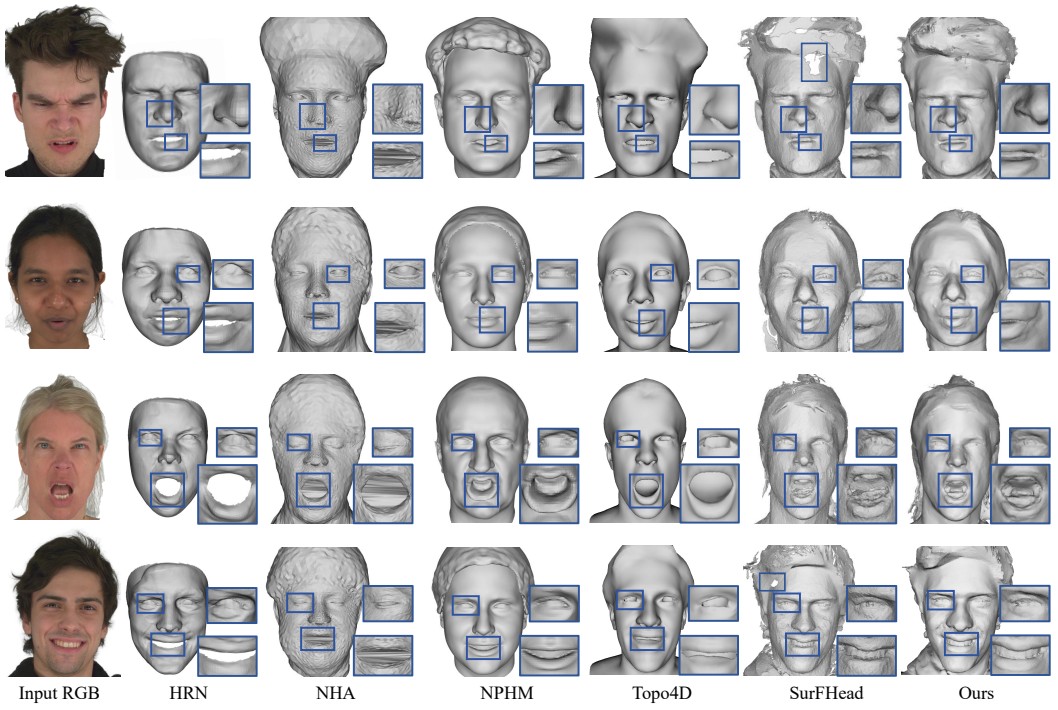

Input RGB    HRN    NHA    NPHM    Topo4D    SurFHead    Ours

Figure 5: Comparison of reconstruction quality from NeRSemble [33] dataset against our baselines.

both "a" and "b", the BVH tree achieves branch rotation (swapping between "b" and "c") to reorganize the hierarchy and do topological simulation. It is important to emphasize that rather than balancing the tree, these rotations are employed to simulate Gaussian movement across volumes, minimizing the overall bounding extent cost—sometimes by introducing **imbalance** into the tree structure. Specifically, we implement these binary-tree rotations inspired by [64].

**Branch Rotations.** On each frame, the BVH tree is updated by a post-order traversal, see Sec. B.1 for details. For rotations, see part (d) of Fig. 2, which illustrates candidate node swaps considered in pivoting. Lift Rebalancer (red rotation) occurs between a parent node's child and grandchild, adjusting subtree hierarchy levels to optimize local structure. Reorder Rebalancer (blue rotation) works among siblings to bypass local optima and achieve better global optimization. Each rotation operation is assigned a cost to filter hopping Gaussians. The green rotation produces mirrored-equivalent trees from Reorder Rotation and is omitted to avoid redundant computations.

**Handling Hopping Points.** When the rotation cost of the middle level nodes exceeds a threshold, the Gaussians within exhibit strong mobility and are filtered as hopping points, and will be incrementally 3D triangulated.

## 4 Experiments

### 4.1 Experiments Settings

**Implementation Details.** To initialize TGA, we adopt VHAP [58] to preprocess the multi-view RGB video dataset for head tracking. We compare our approach against state-of-the-art avatar reconstruction methods via both qualitative and quantitative experiments. Specifically, we evaluate TGA on Chamfer distance, normal error, and recall [65] using the Multiface dataset [66], which provides 3D ground truth. We further evaluate the performance in terms of mesh extraction time, mesh rendering quality, and Gaussian-based novel view synthesis and self-reenactment quality. Thanks to our perspective-correct ray tracing for precise evaluation of Gaussian contributions, TGA converges within 300k iterations. All experiments are performed on NVIDIA RTX 4090 GPUs, using the same hyperparameters as GaussianAvatars [2].

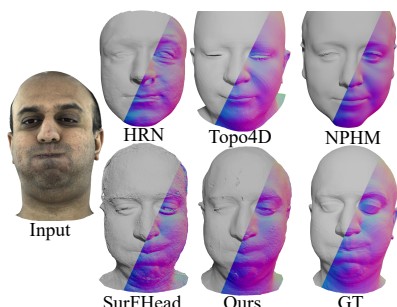

| Method | $L_1$-CD$\downarrow$ | MAE$\downarrow$ | Recall@2.5mm$\uparrow$ |
|---|---|---|---|
| HRN [67] | 2.64 | 22.3 | 0.698 |
| 3DDFA [68] | 4.35 | 22.9 | 0.649 |
| NHA [42] | 6.02 | 28.9 | 0.462 |
| NPHM [24] | 3.35 | 20.5 | 0.764 |
| SF [4] | 2.50 | 24.8 | 0.751 |
| Topo4D [54] | 2.33 | 19.3 | 0.772 |
| **Ours** | **2.16** | **17.7** | **0.802** |

Figure 6: Comparison of reconstructed meshes and normals on the Multiface [66]. Although frontal-view differences appear minor, 3D error metrics show that TGA remains the closest to the GT.

**Memory Overheads.** The memory overheads of training and mesh extraction is no more than 24 gigabytes since we load images on-the-fly. When scaling to longer frame sequences and with stable number of Gaussians, the training memory usage remains mostly constant, and the storage for BVH tree pivoting is related to the number of Gaussians since it is conducted frame by frame.

**Datasets.** We evaluate our method on the NeRSemble [33], Multiface [66] and NHA Dataset [42]. The NeRSemble captures detailed facial dynamics, and the data is calibrated with sub-millimeter accurate camera poses and high-quality foreground segmentation. The Multiface dataset captures subjects covering dense multiview camera captures, rich facial expressions, and ground truth mesh to evaluate the 3D reconstruction efficiency. The NHA real dataset contains sequences that are suitable for the evaluation of full dynamic head approaches. We use it to evaluate the novel-view synthesis and self-reenactment rendering performance of our method.

## 4.2 Baselines

HRN [67] is a hierarchical representation network that achieves detailed face reconstruction. It generates a displacement map from each view and fuse them to obtain the final mesh. 3DDFA-V3 [68] uses geometric guidance for facial part segmentation for face reconstruction. We reconstruct a mesh for each view and then fuse them to obtain a multi-view-consistent final mesh. NHA [42] is learned from a monocular RGB portrait video that features a range of different expressions and views. NPHM [24] generates a signed distance field of a human head given an identity code and an expression code, and can then be translated into a mesh via marching cubes [69]. SurFhead [4] employs 2DGS [70] and GaussianAvatars[2] to reconstruct photorealistic avatars and high-fidelity surface normals and meshes from videos. Topo4D [54] introduces novel texture regularization.

## 4.3 Avatar Reconstruction Results

**Mesh Geometry.** We compare TGA against our baselines on the NeRSemble dataset [33] by reconstructing each avatar from 16-view RGB sequences and present qualitative results in Fig. 5. Comparative experimental results demonstrate that TGA can faithfully capture facial shape and expression details, greatly aiding avatar identity recognition and accurate emotion interpretation. Although HRN [67] reconstructs detailed wrinkle patterns, it still misses personalized eye and nose features. 3DDFA-V3 [68] exhibits jaw-shape inaccuracies. NHA [42] employs high-capacity neural networks for photorealistic rendering, but its geometry remains underconstrained. NPHM [24] excels at expression representation yet fails to preserve identity. Topo4D [54] and SurFHead [4] achieve strong identity reconstruction; however, Topo4D lacks fine facial detail, and SurFHead is constrained by depth maps rendered from 3DGS.

Additionally, we conduct a qualitative and quantitative evaluation of the reconstructed meshes on the Multiface Dataset [66] in Fig. 6. We focus on three metrics: $L_1$-Chamfer distance, normal MAE (Mean Angular Error), and Recall [65] which measures the percentage of ground-truth points within a 2.5 mm threshold of any reconstructed point. TGA consistently produces meshes that closely match the ground truth from both frontal and side viewpoints.

**Mesh Rendering.** Moreover, we evaluate the mesh-based rendering results on the four subjects (shown in Fig. 5) from the frontal view in Tab. 1. Since TGA currently does not incorporate intrinsic

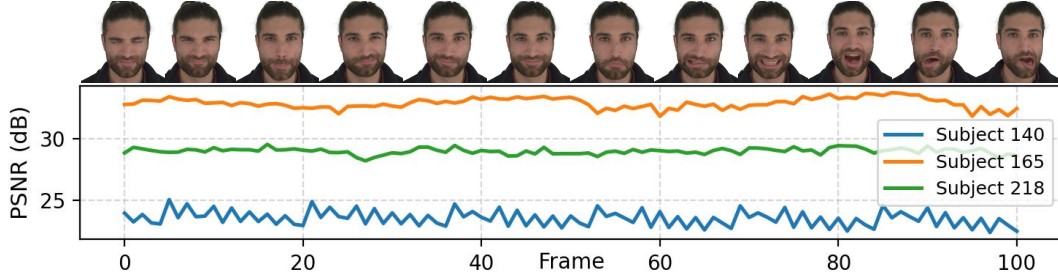

Figure 8: PSNR curves of novel view synthesis results on subjects from the NeRSemble [33].

decomposition or reflectance modeling, we employ flat shading based solely on vertex colors for rendering. We additionally evaluate the temporal quality and consistency of our results on temporal using VMAF [71], a metric designed to capture both perceptual quality and temporal coherence. The qualitative mesh rendering results are represented in the appendix C.2.

**Gaussians Rendering.** Although our method primarily targets geometry-accurate 3D reconstruction, TGA also exhibits strong performance in self-reenactment and novel-view synthesis. We present the comparison with SurFhead [4] and GHA [51] in Fig. 7 and Tab. 1, respectively. The self-reenactment results indicate that, by leveraging the Perspective-Aware Gaussian Transformation, the proposed method can achieve more faithful rendering of dynamic regions (eyes and mouth) with fewer iterations than other 3DGS-based avatar methods. Additionally, we present sequential novel view syn-

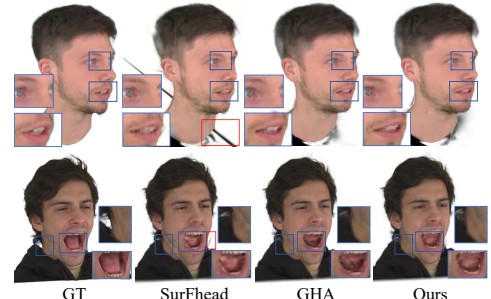

GT    SurFhead    GHA    Ours

Figure 7: Self-reenactment on the NHA [42] and NeRSemble [33] dataset.

thesis rendering frames and their corresponding PSNR curves. The Fig. 8 demonstrates the stability and temporal consistency of TGA.

**Inference Time.** Furthermore, we benchmark mesh-extraction times against baseline methods in Tab. 1. For Gaussian-based methods, meshes are extracted via Truncated Signed Distance Function fusion on 3DGS-rendered depth maps following 2DGS [70]. To ensure fairness, we selected sequences of 100 frames from the Multiface Dataset [66], using 16 key views for tracking and resizing images to the resolution employed by NeRSemble [33]. HRN [67], 3DDFA [68], and NPHM [24] takes approximately 2–5 minutes per frame to inference.

Table 1: Average mesh extraction time, mesh-based rendering and NVS rendering performance.

| Method | Mesh-based Rendering | | | | Gaussian-based Rendering | | | | Inference↓ |
|---|---|---|---|---|---|---|---|---|---|
| | PSNR↑ | SSIM↑ | LPIPS↓ | VMAF [71]↑ | PSNR↑ | SSIM↑ | LPIPS↓ | VMAF [71]↑ | |
| GA [2] | 18.63 | 0.542 | 0.496 | 36.7 | 30.29 | 0.934 | 0.066 | 51.2 | ∼20s |
| Topo4D [54] | 23.69 | 0.637 | **0.285** | **53.9** | 30.88 | 0.931 | 0.064 | 61.4 | While Train |
| SF [4] | 21.09 | 0.558 | 0.479 | 39.1 | 29.94 | 0.933 | 0.062 | 43.6 | ∼20s |
| **Ours** | **23.77** | **0.645** | 0.311 | 47.2 | **31.32** | **0.936** | **0.058** | **65.8** | **∼8s** |

## 4.4 Ablation Study

**Perspective-aware Gaussian Transformation (PGT).** We demonstrate the geometric (CD, MAE and Recall for extracted meshes) and chromatic (PSNR, SSIM, LPIPS for Gaussian-based rendering) impact of the perspective-aware Gaus-

Table 2: Ablations on each modules in the PGT.

| Method | $L_1$-CD ↓ | MAE ↓ | Recall@2.5mm ↑ | PSNR↑ | SSIM↑ | LPIPS↓ |
|---|---|---|---|---|---|---|
| Vanilla | 3.67 | 24.9 | 0.547 | 29.85 | 0.931 | 0.073 |
| w/o Homogeneous | 3.32 | 22.1 | 0.619 | 30.08 | 0.931 | 0.070 |
| w/o Jacobian | 2.58 | 20.1 | 0.748 | 30.85 | 0.934 | 0.066 |
| w/o $L_{nr}$ | 2.43 | 19.2 | 0.756 | 31.17 | 0.935 | 0.061 |
| Ours | **2.16** | **17.7** | **0.802** | **31.32** | **0.936** | **0.058** |

sian transformation in Tab. 2 and Fig. 9. First, the homogeneous formulation yields a noticeable improvement in mesh quality. Second, the Jacobian gradient is essential for guiding deformation of canonical Gaussians in dynamic regions, such as the eyes (red box in Fig. 9). Without the transformed normal loss, Jacobian guided deformation fails to propagate effectively into the geometry, as evidenced by the results in Tab. 2.

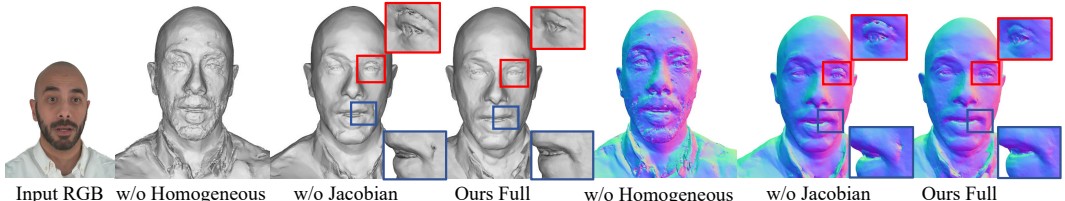

Input RGB   w/o Homogeneous   w/o Jacobian   Ours Full   w/o Homogeneous   w/o Jacobian   Ours Full

Figure 9: The close-ups, taken from $+60°$ view, highlight the effect of our PGT module.

**Hopping Points Filtering.** Furthermore, we evaluate hopping-point filtering via our BVH pivoting mechanism in Fig. 10, rather than relying on a fixed $\Delta$ threshold on Gaussian centers across frames. When filtering with a fixed $\Delta$ Gaussian center between frames, the blue points illustrate (a) incomplete culling and (b) excess screening. In contrast, our BVH pivoting accurately filters out the hopping Gaussians (c). It should be noted that when the avatar slightly opens the eyes or mouth, the forehead topology remains largely unchanged, obviating the need for re-triangulation in that region.

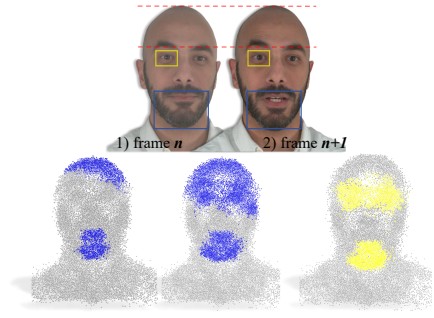

a) Fixed $\Delta = 0.0039$  b) Fixed $\Delta = 0.0034$   c) Ours Hopping

Figure 10: Effect of hopping Gaussians filtering.

**Accelerated Mesh Extraction.** We also evaluate mesh extraction speed across different training iterations by reporting the number of mesh vertices at key checkpoints and the time required for mesh extraction per frame. As training progresses, although the scene contains more points, the Gaussian BVH tree has learned a better topology of the avatar face. As shown in Tab. 4, while the total triangulation time (3rd column) increases with $O(n \log n)$ complexity, our method effectively reduces the computational burden (4th column).

Table 3: Effect of Incremental Strategy for Fast Avatar Reconstruction.

| Iteration | Vertex Num | 1st Frame | $x$-th Frame |
|-----------|------------|-----------|--------------|
| 50k | 258k | 12s | **1.9s** |
| 100k | 372k | 26s | **3.4s** |
| 200k | 489k | 49s | **5.7s** |
| 300k | 563k | 61s | **8.4s** |

## 4.5   Discussion

**Limitations.** Our method struggles in regions like hair and eyes, where translucency, non-rigid motion, and strong specular reflections violate modeling assumptions, leading to opacity inconsistency and hollow-eye artifacts. It also fails under severe expression deformations, causing mesh tearing or topology distortion, since the BVH-hopping Gaussians assume smooth motion. Performing a fresh triangulation can effectively remove accumulated errors.

**Societal impact.** Our method advances high-fidelity facial reconstruction, but it also poses potential risks of misuse, such as identity theft, unauthorized avatar replication, or deepfake generation. These concerns call for thoughtful reflection on ethical implications and the adoption of practical safeguards to minimize possible harm. But with proper and responsible use, we believe our method can bring significant benefits to applications in virtual/augmented reality.

## 5   Conclusion

In this work, we proposed *TGA*, a perspective-aware 4D Gaussian avatar framework that captures fine-grained facial geometry under dynamic conditions. By integrating Jacobian-guided deformation into a homogeneous projection, our method preserves color consistency and geometric accuracy. Coupled with an efficient BVH Tree Pivoting strategy for incremental mesh extraction, TGA achieves state-of-the-art performance in dynamic avatar reconstruction. Future work will explore its extension to relighting, full-body modeling and real-time applications.

# 6 Acknowledgment

This study is partially supported by the National Natural Science Foundation of China under Grants Nos. 62302024, 52441202 and 62202010, the Beijing Nova Program (No.20250484786), the Beijing Advanced Innovation Center for Future Blockchain and Privacy Computing, and the Fundamental Research Funds for the Central Universities.

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

# A Theoretical Analysis

## A.1 Homogeneous Projection and Perspective-Aware Transformation.

Let $C(\mathbf{p}) \in \mathbb{R}^3$ be the rendered color at pixel $\mathbf{p} = (x, y)$ obtained by compositing of $K$ Gaussians,

$$C(\mathbf{p}) = \sum_{k=1}^{K} c_k A_k(\mathbf{p}) \prod_{j<k}(1 - A_j(\mathbf{p})), \quad A_k = \alpha_k D_k, \ D_k = \exp\left[-\tfrac{1}{2}\phi_k^{*2}\right],$$

where $\phi_k^*$ denotes the normalized projected deviation of Gaussian $k$ in image space. Denote the full perspective Jacobian $\mathbf{J}_{\text{persp}} \in \mathbb{R}^{2\times 3}$ by

$$\mathbf{J}_{\text{persp}} = \begin{pmatrix} \frac{f}{Z} & 0 & -\frac{fX}{Z^2} \\ 0 & \frac{f}{Z} & -\frac{fY}{Z^2} \end{pmatrix}$$

and the affine approximation by $\mathbf{J}_{\text{aff}} = \text{diag}(f/Z, f/Z, 0)$ that drops the shear terms. Let $\nabla_p C$ and $\tilde{\nabla}_p C$ be the image-space gradients computed with $\mathbf{J}_{\text{persp}}$ and $\mathbf{J}_{\text{aff}}$, respectively. Then, for any camera pose where at least one off-diagonal entry of $\mathbf{J}_{\text{persp}}$ is non-zero (i.e. any non-frontal view):

$$\left\|\nabla_p C\right\|_2^2 = \left\|\tilde{\nabla}_p C\right\|_2^2 + \underbrace{\sum_{k=1}^{K}\left\|\mathbf{S}_k\,\mathbf{J}_\Delta\right\|_F^2}_{> \, 0} \tag{4}$$

where $\mathbf{J}_\Delta = \mathbf{J}_{\text{persp}} - \mathbf{J}_{\text{aff}}$ contains only the shear components, and $\mathbf{S}_k$ is a symmetric positive matrix that depends on $\alpha_k$, $D_k$, and the transmittance of the front Gaussians. Consequently $\|\nabla_p C\|_2 > \|\tilde{\nabla}_p C\|_2$; i.e. the homogeneous-projection renderer is *strictly more view-sensitive* than its affine counterpart.

*Proof.* For brevity, write $T_{k-1} = \prod_{j<k}(1 - A_j)$ and note that $0 < T_{k-1} \leq 1$. By direct differentiation we have

$$\nabla_p C = \sum_k c_k\, T_{k-1}\left[\nabla_p A_k - A_k \sum_{j<k}\frac{\nabla_p A_j}{1 - A_j}\right], \quad \nabla_p A_k = -\alpha_k D_k \phi_k^* \nabla_p \phi_k^*.$$

All dependence on the projection model is confined to $\nabla_p \phi_k^*$. With either Jacobian choice it can be expressed as a linear map acting on the shear term $\mathbf{J}_\Delta$:

$$\nabla_p \phi_k^* = \mathbf{S}_k\,\mathbf{J}_{\text{persp}}, \quad \tilde{\nabla}_p \phi_k^* = \mathbf{S}_k\,\mathbf{J}_{\text{aff}}, \quad \implies \nabla_p \phi_k^* = \tilde{\nabla}_p \phi_k^* + \mathbf{S}_k\,\mathbf{J}_\Delta.$$

Substituting in $\nabla_p C$ and expanding $\|\nabla_p C\|_2^2$ yields

$$\left\|\nabla_p C\right\|_2^2 = \left\|\tilde{\nabla}_p C\right\|_2^2 + \sum_k\left\|c_k T_{k-1}\alpha_k D_k \phi_k^* \mathbf{S}_k \mathbf{J}_\Delta\right\|_2^2 + \left\langle\tilde{\nabla}_p C, \sum_k c_k T_{k-1}\alpha_k D_k \phi_k^* \mathbf{S}_k \mathbf{J}_\Delta\right\rangle.$$

The cross-term in the inner product vanishes because $\mathbf{J}_\Delta$ contains only off-diagonal elements while $\tilde{\nabla}_p C$ depends exclusively on $\mathbf{J}_{\text{aff}}$ (diagonal). Hence the second equality in (4) follows, and the Frobenius-norm term is strictly positive whenever $\mathbf{J}_\Delta \neq \mathbf{0}$.

**Implication.** Because facial albedo varies weakly, rendering fidelity primarily depends on how strongly the visibility term responds to slight shifts in viewpoint or shape. As shown in Sec. A.1, the homogeneous-projection formulation amplifies this response, leading to higher sensitivity to view changes and thus improved reproduction of fine-scale, view-dependent facial appearance.

# B  Algorithm

## B.1  Pivot and Filter Hopping Points.

Here, we represent the proposed algorithm for Pivot and Filter Hopping Points in Sec. 3.3.2.

---

**Algorithm 1:** Pivot and Filter Hopping Points

---

**Input:** $\mathcal{T}$ (BVH Tree), $\mathcal{G}$ (Gaussians), $\tau$ (Hop Threshold)
**Output:** Updated $\mathcal{T}$, hopping points $\mathcal{H}$

1 **function** PIVOT($n, \mathcal{G}, \mathcal{H}, \tau$)
2     RefitAABB($\mathcal{G}(n)$);
3     **if** *n is leaf* **then**
4         SplitCandidate($n$);
5     **else if** *n has two leaf children* **then**
6         Merge($n$);
7     **else if** $n.depth \geq 3$ **then**
8         **foreach** $\rho \in$ *Lift* & *Reorder Rebalancers* **do**
9             ComputeSAH($n, \rho$);
10         **if** *BestRotation(n)* $\neq$ *NONE* **then**
11             ApplyBestRotation($n$), RefitAABB($\mathcal{G}(n)$), ComputeCost($n, \rho$);
12             **if** *Cost(n)* $> \tau$ **then**
13                 $\mathcal{H} \leftarrow \mathcal{H} \cup \mathcal{G}(n)$;

14 **function** LRD($n, \mathcal{G}, \mathcal{H}, \tau$)
15     **if** $n \neq NULL$ **then**
16         LRD($n$.left, $\mathcal{G}, \mathcal{H}, \tau$), LRD($n$.right, $\mathcal{G}, \mathcal{H}, \tau$);
17         PIVOT($n, \mathcal{G}, \mathcal{H}, \tau$);
18 **foreach** *frame* **do**
19     $\mathcal{H} \leftarrow \emptyset$, LRD($\mathcal{T}, \mathcal{G}, \mathcal{H}, \tau$), IncrementalDelaunay($\mathcal{H}$);

---

# C  Additional Results

We provide more novel view synthesis, self-reenactment, mesh rendering ***sequences*** here.

## C.1  Gaussian Rendering Results.

We represent the novel view synthesis and self-reenactment results on the NeRSemble dataset [33], where one frame is sampled every 5-8 frames for visualization.

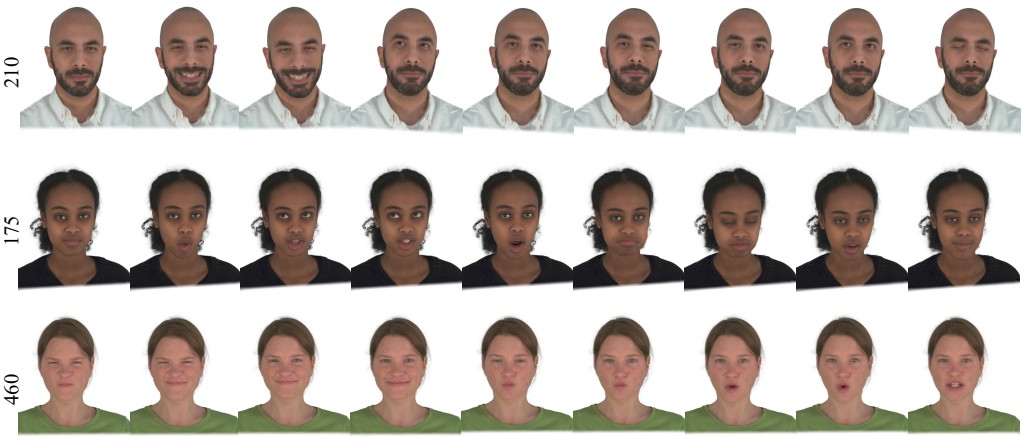

Figure 11: Novel view synthesis rendering results on subjects of the NeRSemble dataset [33].

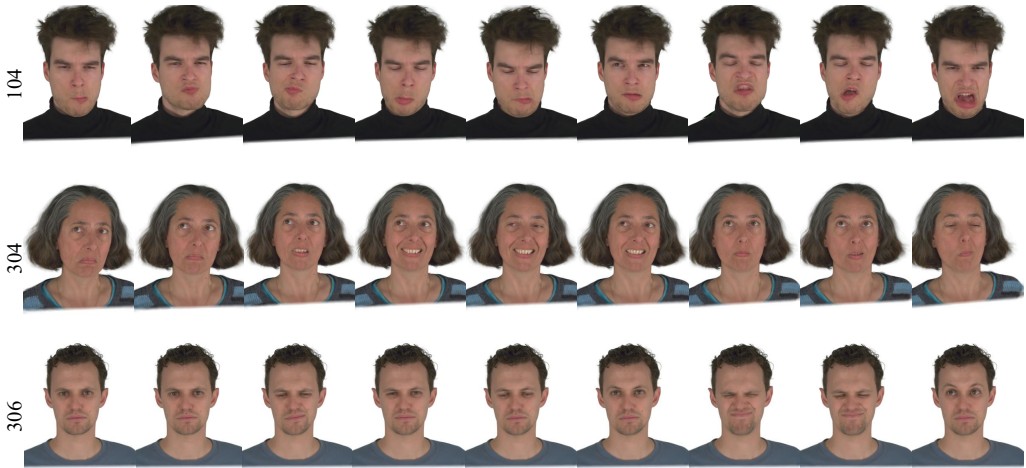

Figure 12: Self-reenactment rendering results on subjects of the NeRSemble dataset [33].

## C.2 Mesh Rendering Results.

Since TGA currently does not incorporate intrinsic decomposition or reflectance modeling, we employ flat shading based on vertex colors for rendering.

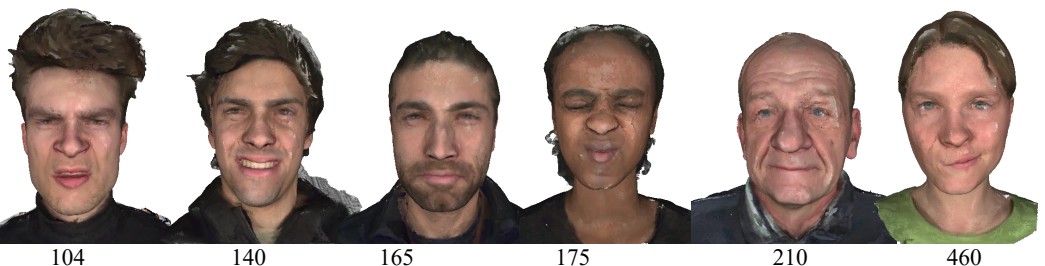

Figure 13: Mesh rendering results by Meshlab on subjects of the NeRSemble dataset [33].

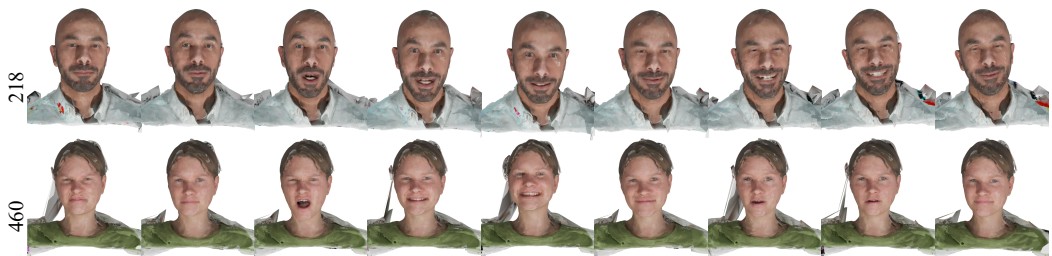

Figure 14: Mesh sequence rendering results by Blender on subjects of the NeRSemble dataset [33].

## D   More Discussion

### D.1   Impact of the focal length.

Shorter focal lengths (i.e., wider fields of view) introduce stronger perspective distortion, making the affine approximation less accurate. Our homogeneous formulation can address this by incorporating full-depth information, enabling more accurate color blending and better geometry reconstruction under wide-FoV conditions.

## D.2 Generalization in sparse-view scenarios.

TGA is primarily designed for multi-view dynamic reconstruction, but it can ensure reasonable results under sparse-view. Specifically, the FLAME model offers strong geometric priors that effectively guide reconstruction in low-coverage settings. The proposed Perspective-aware Transformation leverages subtle photometric changes to optimize Gaussian attributes from limited viewpoints. Additionally, the Jacobian-guided deformation improves inter-frame consistency by enhancing the temporal coverage of Gaussian primitives.

## D.3 Performance on longer videos and longer-term consistency.

Our method can produce relatively stable outputs on longer sequences. When scaling to longer frame sequences and keeping the number of Gaussians stable, the overall training time and rendering metrics are expected to be close to Tab. 1 (since we have to train for 300k iterations), and the runtime of BVH tree pivoting is closely related to the number of Gaussians rather than sequence length.

Specifically, the proposed BVH pivoting mechanism is designed to support long-range consistency. It incrementally tracks topological variations in Gaussian pointcloud and reuses triangulations when no significant local deformation is detected. Additionally, periodic re-triangulation provides a way to reset the accumulated geometric errors and recover accuracy.

# E    More Comparisons

We further compare TGA with existing approaches 3DGUT [72, 73] and 3DGS-marcher [74] about improvement over EWA splatting, modified with FLAME-based mesh-Gaussian deformation. The quantitative results show that our method achieves superior performance in the ***scope of dynamic avatar reconstruction***.

Table 4: Comparisons with existing approaches on improvement over EWA splatting.

| Method | PSNR↑ | SSIM↑ | LPIPS↓ |
|---|---|---|---|
| 3DGUT [72, 73] | 29.96 | 0.938 | 0.084 |
| 3DGS-marcher [74] | 30.54 | 0.941 | 0.079 |
| Ours | 31.37 | 0.952 | 0.058 |

# F    Future Work

Our current work primarily focuses on dynamic reconstruction under controlled indoor lighting and does not explicitly support lighting variations. Lighting changes can indeed affect 3DGS-based methods due to their reliance on photometric consistency. In future work, we plan to incorporate intrinsic decomposition and physically-based rendering into the reconstruction pipeline to better disentangle geometry, reflectance, and illumination. Such integration would enable the model to handle complex and dynamic lighting conditions, thereby extending TGA to more casual and in-the-wild scenarios with improved photometric robustness and relightability.

