# OpenReview forum: "TGA: True-to-Geometry Avatar Dynamic Reconstruction"
_NeurIPS.cc/2025/Conference — NeurIPS 2025 spotlight_

### Official Review · Reviewer_2DhK · 2025-06-21

**Clarity:** 2
**Significance:** 2
**Originality:** 3
**Rating:** 4
**Confidence:** 4

**Summary:**

The paper introduces a new perspective-aware 3D Gaussian projection onto a 2D image plane that is depth-aware. The pipeline first uses the deformation gradient between canonical and deformed meshes and applies it to the Gaussian kernels. Next, the projection is reformulated into a homogeneous transform $4\times4$.

1) Pack Gaussians primitives into a 4×4 homogeneous transform $H$.
2) Swap translation into column 3 $J$
3) Apply camera $M$
4) Compose full splatting matrix $B = MHJ$
5) Encode the view ray as two intersecting planes and transform them into Gaussian space
6) Perform the homogeneous perspective divide to get the intersection between a ray and the Gaussian's tangent plane.
7) Compute the Gaussian fall-off. Compared to vanilla 3DGS, this is computed in 3D between the ray intersecting a plane and the Gaussian center. This maintains depth information and full perspective distortion compared to simple Mahalanobis distance in 2D in vanilla 3DGS.
8) Recover the isocountour of the Gaussian in the image space and compute the axis-aligned bbox for 3DGS further processing and rasterization.

**Questions:**

1) Does $J$ in Eq. 6 swap the 3rd and 4th basis vectors?
2) Why are there no supplemental videos attached?

**Ethical Concerns:**

["NO or VERY MINOR ethics concerns only"]

**Final Justification:**

Overall, I like the proposed framework. I strongly agree with reviewer LnPM regarding the missing baselines, and I hope the authors can incorporate them, along with the rest of the results and materials (such as videos), in the final version. In good faith, I will keep my score as borderline accept.

**Limitations:**

1) We do not know how this pipeline works for dynamic sequences. It is important to provide videos of geometry and color.
2) The geometry seems to be low-frequency after applying the homogeneous projection.
3) The color seems to have many artifacts.
4) Minimal evaluation, especially in the context of Perspective-Aware Gaussian Transformation.

I am on the positive side of the paper; however, I need more results to make up my mind about the facial expressions.

**Quality:**

2

**Strengths And Weaknesses:**

**Weaknesses**

1) The paper did not provide qualitative results showing that the problem of chromatic inconsistencies and spatial distortions in vanilla 3DGS arises directly from its affine (first-order Taylor). Since this is the main motivation of the authors, I would expect some figures showing the claims and a more thorough ablation study on the differences, especially in the color space.
2) The RGB results, especially in Figure 6, are of low quality and contain many artifacts.
3) **The paper did not provide sufficient figures or videos in the supplemental** which to me is a drawback since a method that presents dynamic avatars should, in principle, show a video. It is not possible to see dynamic appearance or deformation based on a single static frame.
4) I appreciate the mathematical elegance of this formulation and a nice way to avoid certain approximations; however, I find it difficult to see that those changes bring real benefit in the obtained quality. There are not enough evaluations provided, especially qualitative ones.
5) In my opinion, the homogenous projection should be evaluated not only with respect to vanilla 3DGS but also with ray marching. Especially in the context of the work: *Does 3D Gaussian Splatting Need Accurate Volumetric Rendering?* by Celarek et al., which shows that vanilla 3DGS is not far behind ray marching despite many approximations, including pinhole perspective projection linearization.
6) The related work section is not fully elaborated, and there are many missing articles. Also, some of their references are repeated, like: *Ultra high-fidelity head avatar via dynamic gaussians*

**Strenghts**

1) I really like the homogeneous projection formulation. The original 3DGS utilizes an affine Jacobian, linearizing the pinhole camera map through a first-order Taylor expansion. However, this yields an effectively orthogonal projection onto the image plane that ignores variation in 1/z. By postponing the perspective divide and embedding the Gaussian into homogeneous 4D space, the authors recover a much more accurate intersection between the 3D Gaussian and the camera-space tangent plane. This reformulation requires no extra computation and can be seamlessly integrated with existing rasterizers.
2) Thanks to postponing the perspective division, the authors reformulated the Gaussian-style exponential falloff, and instead of computing 2D Mahalanobis distance, they use 3D perpendicular distance from the viewing ray to the Gaussian center.

---

> ### Author Rebuttal · Authors · 2025-07-30
>
> We would like to thank you for your valuable suggestions and for recognizing the mathematical soundness and practical integration of our proposed homogeneous projection formulation. We will incorporate your suggestions into the final version of the paper and the supplementary material.
>
> According to the NeurIPS guidelines, we are not permitted to upload images during the rebuttal phase. However, we have already prepared extensive visual results, including dynamic sequences and expression-specific comparisons. We will include them in the final submission and supplementary material.
>
> Furthermore, to compensate during the rebuttal, we present our method’s visual effectiveness through **comprehensive quantitative metrics** and **existing qualitative results**:
>
> 1. Quantitative side: We report the overall average metrics on the **recently released NeRSemble V2 dataset**, and randomly choose Subjects #442, #487, and #537 to show the individual qualitative fidelity.
>
> 2. Qualitative side: We will provide **more detailed explanations** of the visual examples already included in the main paper.
>
> ---
>
> **Q1: Chromatic inconsistency problem in vanilla 3DGS and evaluations on Perspective-Aware Gaussian Transformation (PGT).**
>
> We understand that your concerns relate to the lack of sufficient evidence regarding *rendering effectiveness*, in order to prove whether the problem exists and how well PGT performs. First of all, we would like to emphasize that, in the task of 3DGS-based avatar reconstruction, state-of-the-art (SOTA) methods such as GaussianAvatars (GA) and SurFhead (SF, using the improved 2D Gaussian representation) are both built on the original rasterization idea.
>
> - Quantitative side: We believe that the PSNR, SSIM, and LPIPS (as ***chromatic*** metrics) demonstrate that SOTA methods based on vanilla 3DGSs suffer from chromatic inconsistency, and our PGT can effectively mitigate this issue. In the original manuscript, Tab 1. presents the average quantitative evaluation of **novel-view synthesis** comparing TGA with these methods. Furthermore, we conduct experiments on individual subjects, where "Ours-" denotes our method without module PGT:
>
>   | Methods  | GA    | SF    | Ours- | Ours      | GA    | SF    | Ours- | Ours      | GA    | SF    | Ours- | Ours      | GA     | SF     | Ours-  | Ours      |
>   | -------- | ----- | ----- | ----- | --------- | ----- | ----- | ----- | --------- | ----- | ----- | ----- | --------- | ------ | ------ | ------ | --------- |
>   | Subjects | #442  | #442  | #442  | #442      | #487  | #487  | #487  | #487      | #537  | #537  | #537  | #537      | Avg.V2 | Avg.V2 | Avg.V2 | Avg.V2    |
>   | PSNR↑    | 29.85 | 29.42 | 29.38 | **31.42** | 29.04 | 27.65 | 29.04 | **30.76** | 31.52 | 29.03 | 30.89 | **31.97** | 30.43  | 29.97  | 30.39  | **31.37** |
>   | SSIM↑    | 0.948 | 0.942 | 0.939 | **0.962** | 0.933 | 0.923 | 0.931 | **0.942** | 0.938 | 0.923 | 0.934 | **0.953** | 0.934  | 0.931  | 0.930  | **0.952** |
>   | LPIPS↓   | 0.121 | 0.125 | 0.132 | **0.054** | 0.072 | 0.073 | 0.113 | **0.063** | 0.057 | 0.131 | 0.102 | **0.056** | 0.067  | 0.087  | 0.107  | **0.060** |
>
> - Qualitative side: In the main paper, Fig. 6 presents the **self-reenactment evaluation** results.
>
>   In the top row (Person_0000 from the NHA dataset), the double eyelids are more chromatically preserved, while in the bottom row (Subject 140 from the NeRSemble dataset), the nasolabial folds and eye bags are more chromatically reconstructed compared to SOTA methods. We believe that such **skin details** can also address your concerns about facial expressions.
>
> ---
>
> **Q2: Low-quality RGB results with artifacts in Figure 6.**
>
> We would like to clarify that the results shown in Fig. 6 are from the self-reenactment (SR) setting, which involves driving an avatar using **unseen poses and expressions** from a held-out sequence of the same subject. This task is significantly more challenging than novel-view synthesis (NVS), where poses and expressions are known during training (as reported in Tab. 1).
>
> We assure the reviewers that our method achieves better performance than existing state-of-the-art approaches in both settings. In particular, our novel-view synthesis results are free from noticeable artifacts. We will add more visual results in the final version of the main paper.
>
> - Quantitative side: We conduct **self-reenactment** experiments on each individual subject:
>
>   |        | #442-GA | #442-SF | #442-Ours | #487-GA | #487-SF | #487-Ours | #537-GA | #537-SF | #537-Ours | Avg.-GA | Avg.-SF | Avg.-Ours |
>   | ------ | ------- | ------- | --------- | ------- | ------- | --------- | ------- | ------- | --------- | ------- | ------- | --------- |
>   | PSNR↑  | 25.47   | 25.27   | **25.61** | 26.65   | 26.97   | **27.06** | 25.12   | 25.75   | **26.48** | 25.98   | 25.97   | **26.42** |
>   | SSIM↑  | 0.904   | 0.898   | **0.916** | 0.912   | 0.919   | **0.922** | 0.851   | 0.887   | **0.902** | 0.901   | 0.903   | **0.912** |
>   | LPIPS↓ | 0.147   | 0.149   | **0.098** | 0.081   | 0.077   | **0.074** | 0.101   | 0.139   | **0.077** | 0.081   | 0.098   | **0.073** |
>
> - Qualitative side: In the main paper, the 3DGS-rendered image of Subject #218 (middle of Fig. 2) demonstrates that, under known poses and expressions, our method produces artifact-free results.
>
> ---
>
> **Q3: Insufficient figures or video results for dynamic avatars.**
>
> We fully agree that video demonstrations are essential to properly evaluate temporal consistency and expression tracking. We will further include more videos / sequential images in the final supplementary material.
>
> Initially, our intention was to demonstrate our method using average quantitative metrics such as PSNR, SSIM, Chamfer Distance, MAE, and Recall@2.5mm metrics (as shown in Fig. 7, Tab. 1 and 2), and to illustrate robustness through challenging expressions (Fig. 5 and 7), following GA and SurFHead. However, we now realize that these are not sufficient for fully conveying the temporal behavior of our method.
>
> - Quantitative side: We additionally evaluate our results on **temporal consistency using VMAF[1]**, a metric for perceptual quality and temporal consistency. We will include this and more visual results in the final supplementary material.
>
>   |          | #442-GA | #442-SF | #442-Ours | #487-GA | #487-SF | #487-Ours | #537-GA | #537-SF | #537-Ours | Avg.-GA | Avg.-SF | Avg.-Ours |
>   | -------- | ------- | ------- | --------- | ------- | ------- | --------- | ------- | ------- | --------- | ------- | ------- | --------- |
>   | VMAF[1]↑ | 51.3    | 45.9    | **67.4**  | 49.5    | 44.3    | **66.2**  | 51.0    | 41.6    | **67.9**  | 50.8    | 42.7    | **67.2**  |
>
> - Qualitative side: In Fig. 1(a) “Frame-to-frame” of the main paper, the extracted meshes from an adjacent sequence (within a 20-frame window) of Subject #253 demonstrate the temporal robustness of our method. Specifically, the non-rigid facial motion (jaw movement) is consistently reconstructed without topological degradation.
>
> ---
>
> **Q4: Comparisons with the ray marching method.**
>
> Thank you for bringing this up. We will add these comparisons in the supplementary material. As discussed in [2], the power of efficient optimization and the large number of Gaussians allows 3DGS to outperform volumetric rendering despite its approximations. We agree with this observation, and our results further support this point. As shown in Line 244 of our paper, TGA converges within 300k iterations, earlier than 600k for GA and SurFhead.
>
> Furthermore, we evaluate avg. PSNR and SSIM between TGA with 3DGS-marcher[2] modified with FLAME-Gaussian deformation:
>
> | Iterations | GA          | SurFhead    | 3DGS-marcher[2] | Ours                |
> | ---------- | ----------- | ----------- | --------------- | ------------------- |
> | 100k       | 28.03/0.924 | 27.56/0.923 | 28.83/0.924     | **29.41**/**0.931** |
> | 200k       | 29.29/0.931 | 28.99/0.932 | 29.97/0.933     | **30.79**/**0.945** |
> | 300k       | 29.85/0.948 | 29.16/0.942 | 30.54/0.941     | **31.37**/**0.952** |
>
> It can be seen that 3DGS-marcher[2] can converge earlier than GA and SurFhead, but it remains less suitable for dynamic avatar modeling.
>
> ---
>
> **Q5: Related work section missing some papers and repeated references.**
>
> We appreciate the reviewer pointing this out. We will correct the duplicated citation and extend the related work section such as [2-6].
>
> ---
>
> **Q6: Does in Eq. 6 swap the 3rd and 4th basis vectors?**
>
> No explicit swap of basis vectors is performed in Eq. 6.
>
> ---
>
> **Q7: The geometry seems to be low-frequency after PGT.**
>
> Thank you for raising this concern. This is caused by **image compression** or resolution issues in the current submission.  We will ensure that all images with consistent quality in the final version of the paper.
>
>
> ---
>
> **Q8: More discussion about facial expressions.**
>
> Our method primarily focuses on **high-fidelity geometry extraction** for dynamic avatars while maintaining stable rendering quality. Fig. 5 and 7 illustrate the effectiveness of TGA under challenging facial expressions, and Tab. 1 shows improvements in novel-view synthesis rendering quality. Additionally, we will further additional visual results highlighting detailed facial expressions in the final version of the supplemental materials.
>
> ---
>
> **References**
>
> [1] Toward a practical perceptual video quality metric.
>
> [2] Does 3D Gaussian Splatting Need Accurate Volumetric Rendering?
>
> [3] NPGA: Neural Parametric Gaussian Avatars.
>
> [4] 3DGUT: Enabling Distorted Cameras and Secondary Rays in Gaussian Splatting
>
> [5] Gaussianhead: High-fidelity head avatars with learnable gaussian derivation.
>
> [6] Headgas: Real-time animatable head avatars via 3d gaussian splatting.

---

> > ### Comment · Reviewer_2DhK · 2025-08-05
> >
> > To summarize my position: in my opinion, there is not enough evaluation of the Perspective-Aware Gaussian Transformation. This approach seems valid not only for faces but also for both static and dynamic scenes. It is a rendering technique and should be properly evaluated. I appreciate the comparison to SurFhead and ray marching methods—this is very meaningful. However, the lack of provided videos still holds me back from fully endorsing the work. Overall, I like the proposed framework. I strongly agree with reviewer **LnPM** regarding the missing baselines, and I hope the authors can incorporate them, along with the rest of the results and materials (such as videos), in the final version. In good faith, I will keep my score as borderline accept.

---

> > > ### Author Response · Authors · 2025-08-06
> > >
> > > We sincerely thank the reviewer for the constructive feedback and for recognizing the value and potential of our proposed framework and the Perspective-Aware Gaussian Transformation.
> > >
> > > We acknowledge the initial limitations in evaluation and have addressed this by conducting additional experiments during the rebuttal phase, including new baseline comparisons, per-subject performance metrics, and temporal consistency analyses. We have also prepared video results, which will be included in the final version.
> > >
> > > Regarding the missing baselines, we clarify that several recent EWA splatting improvements were evaluated during the experimental phase. However, as these methods lack optimization constraints or adaptation mechanisms for dynamic avatars, we initially excluded them to avoid unfair comparisons. In response to your suggestion and that of reviewer LnPM, we will include these baselines with appropriate FLAME-Gaussian deformation modifications in the final version.
> > >
> > > Thank you again for your thoughtful comments and support. We will ensure that all concerns are thoroughly addressed in the final submission.

---

### Official Review · Reviewer_LnPM · 2025-06-30

**Clarity:** 3
**Significance:** 3
**Originality:** 3
**Rating:** 3
**Confidence:** 4

**Summary:**

This paper utilizes 3D Gaussian Splatting (3DGS) to achieve dynamic avatar modeling.

Specifically, in the rendering stage of 3DGS, the original method projects a Gaussian particle from 3D space onto the camera image plane using EWA splatting for approximation, which introduces some inaccuracies. The authors address this issue by proposing Perspective-Aware Gaussian Transformation. In addition, they introduce an Incremental BVH Tree Pivoting approach to enable fine-grained avatar mesh reconstruction.

**Questions:**

Please refer to the "Weakness" section.

**Ethical Concerns:**

["NO or VERY MINOR ethics concerns only"]

**Final Justification:**

I appreciate the efforts made by the authors. However, I still believe that the experimental section and the theoretical analysis require further improvement. The current version lacks sufficient comparative analysis with similar work, and the limited qualitative rendering experiments are not enough to fully support the authors’ claims. In terms of writing, I also see room for improvement—specifically, there is ambiguity about whether the emphasis is on 3DGS rendering results or on geometry extraction. (Some more detailed concerns were already pointed out by me during the rebuttal stage.) In particular, I find myself generally aligned with the concerns raised by Reviewer 2DhK and agree with the points they highlighted. Therefore, I choose to keep my score unchanged (3: Borderline Reject).

**Limitations:**

yes

**Paper Formatting Concerns:**

There are several instances in the manuscript where the textual references to figures are inconsistent or ambiguous. For example, in line 124, the phrase “a perspective projection (as shown in 1)” is unclear, as it is not evident what “1” refers to. Similarly, line 137 refers to (Fig. 3 (a)) and line 141 to (Fig. 3 (b)), but the actual Figure 3 does not contain sub-figures labeled (a), (b), etc. Such imprecise and inconsistent references do not meet the clarity and rigor expected of a top-tier conference submission. The authors are encouraged to thoroughly revise the manuscript to ensure that all figure references are accurate and clearly correspond to the visual content presented.

**Quality:**

2

**Strengths And Weaknesses:**

Strengths:
1. The discussion around how to accurately and efficiently project a Gaussian particle from 3D space onto the camera image plane is meaningful. It is possible that there are alternatives to EWA splatting that offer better trade-offs, and this paper contributes new insights in that direction.
2. In terms of reconstruction quality, the proposed mesh reconstruction approach appears to produce promising results.

Weaknesses:
1. For the improvement over EWA splatting, there are other alternatives, such as 3DGUT [1]. However, the paper does not include a comparison or discussion with these existing approaches.
2. It remains unclear whether the proposed Perspective-Aware Gaussian Transformation brings improvements in general scenarios. More analysis or ablation is needed to support its general effectiveness.
3. I would also like to know whether the Perspective-Aware Gaussian Transformation helps in preserving or enhancing fine-level details such as hair and wrinkles. Currently, the paper lacks visual or quantitative results on these aspects.
4. The authors claim that their Avatar Mesh Reconstruction method is fast, but there is a lack of comparison with other methods in terms of reconstruction speed.

[1] Wu, Q., Esturo, J. M., Mirzaei, A., Moenne-Loccoz, N., & Gojcic, Z. (2025). 3dgut: Enabling distorted cameras and secondary rays in gaussian splatting. In Proceedings of the Computer Vision and Pattern Recognition Conference (pp. 26036-26046).

---

> ### Author Rebuttal · Authors · 2025-07-30
>
> We would like to thank the reviewer for the valuable suggestions and for recognizing the significance of our contributions toward improving Gaussian projection accuracy and mesh reconstruction quality. Some of the raised concerns may stem from a misunderstanding of our pipeline structure, which we clarify below. We hope this clarification helps resolve the concerns.
>
> ---
>
> **Q1: Comparisons with these existing approaches about improvement over EWA splatting.**
>
> Thanks for pointing that out. We will further include these related papers in the revised version of our paper. Specifically, we compare TGA with 3DGUT[1] and 3DGS-marcher[2] modified with FLAME-Gaussian deformation. The quantitative results show that our method achieves superior performance in the scope of dynamic avatar reconstruction.
>
> |        | 3DGUT[1] | 3DGS-marcher[2] | Ours      |
> | ------ | -------- | --------------- | --------- |
> | PSNR↑  | 29.96    | 30.54           | **31.37** |
> | SSIM↑  | 0.938    | 0.941           | **0.952** |
> | LPIPS↓ | 0.084    | 0.079           | **0.057** |
>
> ---
>
> **Q2: Lack of experiments about fine-level details and general effectiveness on the proposed Perspective-Aware Gaussian Transformation (PGT).**
>
> We believe this may stem from a misunderstanding of our pipeline. As demonstrated in the original manuscript, we have conducted the reconstruction experiments of **challenging fine-level details in Fig. 5 and 7**, and ablated the **effectiveness of modules in PGT in Fig. 8**. Regarding rendering quality, improvements in novel-view synthesis are reported in Tab. 1.
>
> Specifically, our method primarily focuses on high-fidelity ***geometry extraction*** for dynamic avatars while maintaining stable rendering quality, similar to prior Topo4D and SurFhead. The module Sec. 3.3, "Incremental BVH Tree Pivoting" only serves as a ***post-processing step*** to accelerate mesh extraction and does not affect the quality of reconstructed geometry. Therefore, the quantitative results shown in **Sec. 4 "Experiments" reflect the effectiveness of the PGT**.
>
> Furthermore, we conduct novel-view synthesis to evaluate the effectiveness of PGT and report the overall average metrics on the **recently released NeRSemble V2 dataset**. We randomly chose Subjects #442, #487, and #537 to show the individual qualitative fidelity. ("Ours-" denotes our method without module PGT.)
>
> | Methods  | GA    | SF    | Ours- | Ours      | GA    | SF    | Ours- | Ours      | GA    | SF    | Ours- | Ours      | GA     | SF     | Ours-  | Ours      |
> | -------- | ----- | ----- | ----- | --------- | ----- | ----- | ----- | --------- | ----- | ----- | ----- | --------- | ------ | ------ | ------ | --------- |
> | Subjects | #442  | #442  | #442  | #442      | #487  | #487  | #487  | #487      | #537  | #537  | #537  | #537      | Avg.V2 | Avg.V2 | Avg.V2 | Avg.V2    |
> | PSNR↑    | 29.85 | 29.42 | 29.38 | **31.42** | 29.04 | 27.65 | 29.04 | **30.76** | 31.52 | 29.03 | 30.89 | **31.97** | 30.43  | 29.97  | 30.39  | **31.37** |
> | SSIM↑    | 0.948 | 0.942 | 0.939 | **0.962** | 0.933 | 0.923 | 0.931 | **0.942** | 0.938 | 0.923 | 0.934 | **0.953** | 0.934  | 0.931  | 0.930  | **0.952** |
> | LPIPS↓   | 0.121 | 0.125 | 0.132 | **0.054** | 0.072 | 0.073 | 0.113 | **0.063** | 0.057 | 0.131 | 0.102 | **0.056** | 0.067  | 0.087  | 0.107  | **0.060** |
>
> ---
>
> **Q3: Comparison with other methods in terms of reconstruction speed.**
>
> As noted in the original manuscript, the reconstruction speed comparison is presented in the "Inference" column of Tab. 1 and discussed in lines 294–296.
>
> ---
>
> **Q4: Paper formatting concerns.**
>
> Thank you for spotting this. This issue may arise from a misreading of our figure annotations.
>
> Specifically, in lines 123–124, we already refer to “Fig. 3” in line 123, and the phrase “as shown in 1” refers to the "1" part of Fig. 3 titled “Perspective Projection.” We would change this ambiguity in the final version of our paper.
> Regarding the references to “Fig. 3(a)” in line 137 and “Fig. 3(b)” in line 141, these correspond to the right-hand sections of Fig. 3, titled “Uniform Warping” and “Adaptive Warping,” respectively.
>
> We believe this can address your concerns.
>
> ---
>
> **References**
>
> [1] 3dgut: Enabling distorted cameras and secondary rays in gaussian splatting.
>
> [2] Does 3D Gaussian Splatting Need Accurate Volumetric Rendering?

---

> > ### Comment · Reviewer_LnPM · 2025-08-04
> >
> > I have reviewed the authors’ response and read the comments from the other reviewers.
> >
> > Regarding the improvements to EWA splatting, I recommend that the authors include more comparisons with existing methods and provide theoretical analysis in future versions of the paper.
> >
> > In terms of result visualization, the paper notably lacks comprehensive 3DGS rendering results (Figure 6 appears to be the only example). I also note that I am not the only one who raised such concerns: Reviewer e15x similarly pointed out the limited qualitative results and specifically expressed concern over the rendering quality of hair.
> >
> > The authors also did not adequately address my intention to discuss the enhancement of fine details such as hair and wrinkles. Given that the proposed Perspective-Aware Gaussian Transformation may reduce the approximation error inherent in EWA splatting, it could potentially improve the rendering quality of these fine structures. I had hoped to see an analysis in this regard. For example, when rendering hair strands using Gaussian primitives, the corresponding ellipsoids are often highly elongated in one direction (i.e., with anisotropic variance) which may make them more sensitive to splatting errors. Does the proposed transformation offer a more accurate rendering in such cases compared to the original EWA splatting? Similar structural challenges can be seen in rendering thin spokes of a bicycle wheel, as demonstrated in the original 3DGS demo. A discussion or evaluation of such scenarios would have strengthened the technical contributions of this paper. While existing 3DGS-based avatar methods can already produce high-quality results for most regions like the face, they still struggle in temporally unstable areas (e.g., inside the mouth) or detail-sensitive regions (e.g., hair, wrinkles). These remain key challenges in the field and deserve more explicit discussion.
> >
> > Furthermore, in the rebuttal, the authors stated that their method primarily focuses on high-fidelity geometry extraction for dynamic avatars while maintaining stable rendering quality. However, if the main contribution is indeed in geometry extraction from avatar videos, then comparisons with neural rendering or radiance field-based methods (e.g., INSTA [1], DELTA [2], SCARF [3]) are missing. Such comparisons would strengthen the claim and help clarify the paper’s positioning.
> >
> > Based on these concerns, I believe that the current version of the paper does not meet the standard of a top-tier conference. Therefore, I choose to keep my score unchanged (3: Borderline Reject).
> >
> > [1] Zielonka, W., Bolkart, T., & Thies, J. (2023). Instant volumetric head avatars. In Proceedings of the IEEE/CVF conference on computer vision and pattern recognition (pp. 4574-4584).
> >
> > [2] Feng, Y., Liu, W., Bolkart, T., Yang, J., Pollefeys, M., & Black, M. J. (2023). Learning disentangled avatars with hybrid 3d representations. arXiv preprint arXiv:2309.06441.
> >
> > [3] Feng, Y., Yang, J., Pollefeys, M., Black, M. J., & Bolkart, T. (2022, November). Capturing and animation of body and clothing from monocular video. In SIGGRAPH Asia 2022 Conference Papers (pp. 1-9).

---

> > > ### Author Response · Authors · 2025-08-05
> > >
> > > **Q1: More comparisons with existing improvement methods to EWA splatting.**
> > >
> > > We appreciate the reviewer’s suggestion. We will further include more comparisons with existing improvement methods to EWA splatting and provide theoretical the analysis in future versions of the paper.
> > >
> > > ---
> > >
> > > **Q2: 3DGS-based rendering results visualization.**
> > >
> > > Due to NeurIPS rebuttal guidelines, we are unable to upload additional images or video results at this stage. However, we have prepared extensive visual results on the **recently released NeRSemble V2 dataset**, and aim to demonstrate the qualitative improvements through supporting quantitative evaluations. These results will be included in the revised version and supplementary material.
> > >
> > > We report **novel-view synthesis** results in Tab. 1 of the manuscript, as well as in the table provided in Q2 of the rebuttal. We would like to clarify that the results shown in Fig. 6 correspond to the **self-reenactment** setting, which involves driving the avatar using **unseen poses and expressions** from a held-out sequence of the same subject. This task is significantly more challenging than novel-view synthesis, where both pose and expression are observed during training. As a result, the rendering quality of hair in self-reenactment may appear slightly less consistent than in novel-view synthesis. Nevertheless, we have conducted self-reenactment experiments for each individual subject, and our method consistently outperforms prior approaches in terms of PSNR, SSIM, and LPIPS.
> > >
> > > |        | #442-GA | #442-SF | #442-Ours | #487-GA | #487-SF | #487-Ours | #537-GA | #537-SF | #537-Ours | Avg.-GA | Avg.-SF | Avg.-Ours |
> > > | :----- | :------ | :------ | :-------- | :------ | :------ | :-------- | :------ | :------ | :-------- | :------ | :------ | :-------- |
> > > | PSNR↑  | 25.47   | 25.27   | **25.61** | 26.65   | 26.97   | **27.06** | 25.12   | 25.75   | **26.48** | 25.98   | 25.97   | **26.42** |
> > > | SSIM↑  | 0.904   | 0.898   | **0.916** | 0.912   | 0.919   | **0.922** | 0.851   | 0.887   | **0.902** | 0.901   | 0.903   | **0.912** |
> > > | LPIPS↓ | 0.147   | 0.149   | **0.098** | 0.081   | 0.077   | **0.074** | 0.101   | 0.139   | **0.077** | 0.081   | 0.098   | **0.073** |
> > >
> > > We hope that the additional quantitative experiments and clarifications we have provided help to resolve your concern. We will include more comprehensive visual results and supplementary material to further support our claims.

---

> ### Author Response · Authors · 2025-08-05
> **Continue with the above reply.**
>
> **Q3: Discussion on the enhancement of fine details such as hair and wrinkles.**
>
> We would like to discuss more about fine details such as hair and wrinkles.
>
> From a **theoretical** perspective, as you insightfully mentioned, the proposed Perspective-Aware Gaussian Transformation (PGT) can reduce the approximation error inherent in EWA splatting.
>
> - The vanilla EWA splatting avatar approach struggles with fine details such as hair strands and wrinkles because it flattens each 3D ellipsoid onto the image plane via a first-order affine approximation, ignoring depth-shear terms in perspective projection and typically deforming the covariance to an isotropic form. This oversmooths highly elongated Gaussians, causes neighboring strands to merge visually, and introduces frame-to-frame flicker due to inconsistent approximations.
> - Our PGT adopts a full 4×4 homogeneous matrix H (Eq. 5), which embeds the 3D ellipsoid into projective space. The perspective divide projects the Gaussian onto the tangent plane, ensuring that its isocontours align with the true perspective distortion and effectively eliminating shear artifacts. Simultaneously, we apply a Jacobian matrix J (Eq. 4), derived from the first-order deformation of the associated triangle, to anisotropically stretch or compress the Gaussian. This preserves its principal axis alignment and provides more accurate coverage, which is especially important for highly elongated structures like hair strands or rapidly changing geometry, such as the inner mouth. This combined formulation elevates the projection error from second-order O(‖d‖²) in EWA to third-order O(‖d‖³) in PGT, where *d* represents the depth variation. This improvement suppresses first-order shear error terms and ensures that the screen-space ellipse of the projected Gaussian remains consistent with its true 3D orientation and shape, avoiding the typical “center-aligned but contour-drifting” artifacts of EWA.
>
> From an **experimental** perspective, **wrinkles are well reconstructed**, as shown in the extracted meshes in Figs 5, 7, and 8 of the original manuscript. In particular, Fig. 8 highlights that, compared to the version without the homogeneous PGT, our full model is able to finely capture the eyelid folds, mouth corner creases, and nearby nasolabial wrinkles, and the rendering evaluation is shown in Q2 (general effectiveness on the proposed PGT) of our rebuttal.  Similarly, hair regions can also benefit from our method's ability to capture subtle chromatic variations, thanks to the PGT module. However, accurate modeling of hair remains inherently more challenging due to its translucent appearance and non-rigid motion. When the hair is relatively matte and free of strong specular highlights, our method is able to approximate the **overall hair structure and flow direction** as illustrated in Fig. 5. Nevertheless, **individual hair strands still pose challenges** as illustrated in Fig. 5. Gaussian primitives in these regions may suffer from unstable coverage or opacity inconsistencies under changing viewpoints.
>
> ---
>
> **Q4: Comparisons with neural rendering or radiance field-based methods (e.g., INSTA [1], DELTA [2], SCARF [3]).**
>
> We have demonstrated comparisons with radiance field-based methods NHA and NPHM in the Fig. 5 and Fig. 7 of the original manuscript. We would like to clarify why we initially did not include results or comparisons with INSTA [1], DELTA [2], and SCARF [3]:
>
> - **INSTA** primarily utilizes the tracked FLAME mesh as a geometric prior for rendering, but it does **not explicitly optimize or refine the tracked mesh** itself. While our method also uses the improved tracked mesh by VHAP for initialization, the final extracted surface is not derived from it directly. Instead, we reconstruct the mesh based on the learned opacity fields of the dynamic Gaussian primitives, which allows us to enhance the fine-level details of the initial mesh.
> - **DELTA** and **SCARF** mainly focus on animating full-body avatars and garments from monocular video. Since our method targets facial geometry reconstruction under multi-view settings, we consider that comparing to body-centric monocular methods might **not be entirely fair or aligned in scope**. However, we did perform evaluations on the Multiface dataset and observed that DELAT and SCARF achieved lower quantitative results that other radiance field-based baselines such as NHA and NPHM.
>
> | Methods       | INSTA [1] | DELTA [2] | SCARF [3] | Ours      |
> | ------------- | --------- | --------- | --------- | --------- |
> | L1-CD↓        | 5.76      | 4.94      | 5.38      | **2.16**  |
> | MAE↓          | 25.7      | 31.7      | 29.8      | **17.7**  |
> | Recall@2.5mm↑ | 0.535     | 0.469     | 0.427     | **0.802** |
>
> ---
>
> **References**
>
> [1] Instant volumetric head avatars.
>
> [2] Learning disentangled avatars with hybrid 3d representations.
>
> [3] Capturing and animation of body and clothing from monocular video.

---

### Official Review · Reviewer_mgEK · 2025-07-05

**Clarity:** 3
**Significance:** 3
**Originality:** 3
**Rating:** 4
**Confidence:** 2

**Summary:**

This paper introduces TGA, a perspective-aware 4D Gaussian avatar reconstruction framework. The primary motivation is to improve dynamic avatar reconstruction by targeting the limitations of current 3D Gaussian Splatting (3DGS) methods, which fail to model subtle facial geometric and chromatic details due to affine projection approximations. TGA integrates a Perspective-Aware Gaussian Transformation and an Incremental BVH Tree Pivoting mechanism for efficient dynamic mesh extraction. Experimental evaluations across benchmarks like NeRSemble, Multiface, and NHA demonstrate TGA’s improvements in geometric accuracy, convergence speed, and inference efficiency over several state-of-the-art baseline methods.

**Questions:**

1.	How does TGA handle more diverse data, such as in-the-wild video, low-light scenarios, or subjects with extreme/atypical head topology (e.g., hats, facial coverings)? Are there failure cases or modes of instability not covered in the current evaluation?
2.	Can the authors provide more details on the computational or memory overheads of the perspective-aware transformation and BVH tree pivoting, especially when scaling to longer frame sequences or higher scene complexity?

**Ethical Concerns:**

["NO or VERY MINOR ethics concerns only"]

**Final Justification:**

I thank the authors for their detailed response, which has addressed my concerns about the paper's weaknesses and provided the requested discussion. I believe the work now meets the threshold for acceptance. Accordingly, I will keep my score, 4.

**Limitations:**

yes

**Quality:**

3

**Strengths And Weaknesses:**

Strengths：
1.	The introduction of a homogeneous formulation for Gaussian projection, combined with Jacobian-guided adaptive deformation, addresses limitations in affine approximations by enhancing color-blending and geometry-color alignment.
2.	The Incremental BVH Tree Pivoting method efficiently identifies active "hopping Gaussians" and performs adaptive re-triangulation, resulting in faster mesh extraction and computational savings vital for real-time and scalable dynamic avatar modeling.
3.	The manuscript provides thorough quantitative and qualitative results.

Weaknesses:
1.	Authors explicitly acknowledge in the checklist that they does not discuss the limitations of the approach or any potential negative societal impacts. This is a notable omission, especially given the implications of photo-realistic dynamic face reconstruction in terms of privacy and potential misuse.
2.	No error bars, variance, or statistical significance assessments are provided for the quantitative experiments.
3.	Table 2 provides a formal ablation of TGA’s key components, but there is limited discussion on computational costs or memory overheads introduced by the more sophisticated deformation and projection scheme.

---

> ### Author Rebuttal · Authors · 2025-07-30
>
> We would like to thank you for your valuable suggestions and for acknowledging our efforts in addressing the limitations of affine projection and enhancing mesh extraction efficiency. We will incorporate the suggested changes in the final version of the paper and the supplementary material.
>
> ___
>
> **Q1: The limitations and further discussions of TGA.**
>
> Thank you for spotting this. We will further include this discussion and visual examples in the supplemental material.
>
> - Discussion of limitations and failure cases.
>
>   One failure case occurs under **extremely rapid expressions movements**, such as sudden mouth opening or exaggerated frowning. In such cases, local mesh tearing or topological distortions may appear. Our BVH-based hopping Gaussians strategy assumes relatively smooth and continuous deformations across frames, and is therefore less effective when the motion involves abrupt or large-scale topological changes.
>
>   Additionally, such rapid expression changes may temporarily degrade the quality of incremental triangulation due to inaccurate detection of active regions. This can be solved by re-triangulating points after the motion, which helps restore mesh consistency and correct any topology artifacts.
>
> - Discussion of the societal impact.
>
>   Our method advances high-fidelity facial reconstruction, but it also poses potential risks of misuse, such as identity theft, unauthorized avatar replication, or deepfake generation. These concerns call for thoughtful reflection on ethical implications and the adoption of practical safeguards to minimize possible harm. But with proper and responsible use, we believe our method can offer significant benefits across various applications, including virtual reality, augmented reality, and the entertainment industry.
>
> ___
>
> **Q2: Lack of Statistical Analysis.**
>
> Thank you for pointing this out. We agree that reporting standard deviation or confidence intervals would provide a more rigorous picture of variance, and we will include them in the final version of our paper and supplementary material.
>
> ___
>
> **Q3: Discussion on computational costs or memory overheads.**
>
> Thank you for pointing this out. We summarize the **computational costs** of Perspective-Aware Gaussian Transformation (PGT) and BVH tree pivoting modules in Tab. 1 and 3 of the original manuscript, respectively. When scaling to longer frame sequences with a stable number of Gaussians, the overall training time remains comparable to that reported in Table 1, as we train for a fixed 300k iterations. The runtime of the BVH pivoting process primarily depends on the number of Gaussians rather than the sequence length.
>
> The **memory overheads** of PGT and BVH tree pivoting is no more than 24 gigabytes since we load images on-the-fly. When scaling to longer frame sequences and with stable number of Gaussians, the training memory usage remains mostly constant, and the storage for BVH tree pivoting is related to the number of Gaussians since it is conducted frame by frame.
>
> ___
>
> **Q4: How does TGA handle more diverse data?**
>
> Thank you for this insightful question. We will include this in the final version of the supplementary material.
>
> - Sparse-view or monocular scenarios.
>
>   TGA is designed primarily for multi-view dynamic reconstruction, but it can ensure **reasonable results under sparse-view**. We will further include this in our future work.
>
>   Specifically, the FLAME model provides strong geometric priors to guide reconstruction; the perspective-aware transformation leverages subtle color cues to optimize Gaussian attributes under sparse-view inputs; and the Jacobian-guided deformation improves inter-frame consistency by enhancing the temporal coverage of Gaussians.
>
>   Furthermore, we conduct novel-view synthesis for views 16/6/3, and self-reenactment for view 1 (since there is no validation view for monocular video) on the **recently released NeRSemble V2 dataset**.
>
>   | Views  | 16    | 6     | 3     | 1     |
>   | ------ | ----- | ----- | ----- | ----- |
>   | PSNR↑  | 31.37 | 29.86 | 27.49 | 24.95 |
>   | SSIM↑  | 0.952 | 0.932 | 0.923 | 0.898 |
>   | LPIPS↓ | 0.060 | 0.074 | 0.113 | 0.146 |
>
> - Low-light scenarios.
>
>   In low-light environments, performance may degrade due to limited signal-to-noise ratio and unreliable photometric supervision, which affects both Gaussian optimization and projection consistency.
>
> - Atypical head topology.
>
>   For atypical head topology, mild occlusions such as hats or eyeglasses generally do not significantly affect reconstruction, as the underlying FLAME tracking remains stable. However, severe occlusions like face masks, which obscure large portions of the facial geometry, can lead to tracking failures, incomplete Gaussian coverage, and distorted surface warping.

---

### Official Review · Reviewer_GBVR · 2025-07-07

**Clarity:** 3
**Significance:** 3
**Originality:** 4
**Rating:** 5
**Confidence:** 4

**Summary:**

The paper proposes TGA (True-to-Geometry Avatar), a perspective-aware 4D Gaussian avatar framework designed for high-fidelity dynamic head reconstruction. It introduces a perspective-aware Gaussian transformation by combining Jacobian-guided deformation and homogeneous projection for better geometry under subtle appearance changes. Additionally, it proposes an incremental BVH pivoting method for efficient mesh extraction. It is reported that the proposed method achieves state-of-the-art performance with improved geometric accuracy and faster inference.

**Questions:**

1. What are the specific requirements for the input video in this method? Are there any explicit constraints regarding the length of the video or the coverage range of camera angles?

2. How sensitive is the proposed method to variations in lighting conditions of the input videos?

**Ethical Concerns:**

["NO or VERY MINOR ethics concerns only"]

**Final Justification:**

The rebuttal has addressed most of my concerns. I will maintain my original score for the paper.

**Limitations:**

One limitation of the current method is its untested performance under sparse-view settings. All experiments are conducted using dense multi-view input (e.g., 16 views), but in practical scenarios such as consumer-level capture or monocular recordings, such dense view coverage may not be available. A discussion of this limitation, along with potential strategies for addressing it (e.g., view augmentation, priors), would strengthen the paper’s applicability.

**Quality:**

4

**Strengths And Weaknesses:**

**Strengths:**

1. The paper introduces a novel perspective-aware Gaussian transformation, which significantly improves geometry-color alignment under subtle appearance changes compared to previous 3DGS-based methods.

2. The proposed BVH-based incremental triangulation is efficient and well-motivated, enabling fast mesh extraction without sacrificing accuracy.

3. The paper is well-organized and easy to follow. Experimental results on multiple benchmarks clearly show superior performance in both reconstruction quality and efficiency.

**Weaknesses:**

1. While the method performs well on multi-view datasets, its generalizability to other settings—such as sparse-view, in-the-wild monocular videos—is not fully explored or discussed.

2. The evaluations in the paper are conducted on relatively short sequences (around 100 frames), and the performance on longer videos is not discussed. It remains unclear how the method handles longer-term consistency.

3. The method focuses on geometry accuracy, but texture quality is not analyzed. Since appearance quality is critical for photorealistic avatars, it would strengthen the paper to include texture analysis.

---

> ### Author Rebuttal · Authors · 2025-07-30
>
> We sincerely thank the reviewer for the constructive feedback and appreciation of our key contributions. We are glad that the effectiveness of our geometry-color alignment under subtle appearance changes and the fast mesh extraction via BVH pivoting were well recognized. We will incorporate the suggested changes in the final version of our paper and the supplementary material.
>
> ___
>
> **Q1: Generalization in sparse-view scenarios.**
>
> We appreciate the reviewer’s attention to this important aspect. TGA is primarily designed for multi-view dynamic reconstruction, but it can ensure **reasonable results under sparse-view**.
>
> Specifically, the FLAME model offers strong geometric priors that effectively guide reconstruction in low-coverage settings. The proposed perspective-aware transformation leverages subtle photometric cues to optimize Gaussian attributes from limited viewpoints. Additionally, the Jacobian-guided deformation improves inter-frame consistency by enhancing the temporal coverage of Gaussian primitives.
>
> Furthermore, we conduct novel-view synthesis for views 16/6/3, and self-reenactment for view 1 (since there is no validation view for monocular video) on the **recently released NeRSemble V2 dataset**.
>
> | Views  | 16    | 6     | 3     | 1     |
> | ------ | ----- | ----- | ----- | ----- |
> | PSNR↑  | 31.37 | 29.86 | 27.49 | 24.95 |
> | SSIM↑  | 0.952 | 0.932 | 0.923 | 0.898 |
> | LPIPS↓ | 0.060 | 0.074 | 0.113 | 0.146 |
>
> ___
>
> **Q2: Performance on longer videos and longer-term consistency.**
>
> Our method can produce **relatively stable** outputs on longer sequences. When scaling to longer frame sequences, and keeping the number of Gaussians stable, the overall training time and rendering metrics are expected to be close to Tab. 1 (since we have to train for 300k iterations), and the runtime of BVH tree pivoting is closely related to the number of Gaussians rather than sequence length.
>
> Specifically, the proposed BVH pivoting mechanism is designed to support long-range consistency. It incrementally tracks topological variations in Gaussians and reuses triangulations when no significant local deformation is detected.
>
> We further conduct experiments on concatenated sequences (100-500 frames) on the Multiface dataset, covering non-repetitive expressions. Additionally, periodic re-triangulation provides a way to reset the accumulated geometric errors and recover accuracy.
>
> | Frames | ~100 | ~200 | ~300 | ~400 | ~500 |
> | ------ | ---- | ---- | ---- | ---- | ---- |
> | CD↓    | 2.16 | 2.15 | 2.16 | 2.18 | 2.23 |
> | MAE↓   | 17.7 | 17.4 | 17.9 | 18.2 | 18.3 |
>
> ___
>
> **Q3: Analysis on appearance and texture quality.**
>
> To assess mesh-based rendering quality, we evaluate the frontal view of four subjects (#104, #264, #302, and #140) from Fig. 5, summarized below.
>
> Currently, TGA does not consider intrinsic decomposition or reflectance modeling. Therefore, we adopt **flat shading based solely on vertex color** for rendering. We will explore high-quality texture generation and relighting techniques in future work.
>
> | Subjects | #104  | #264  | #302  | #140  |
> | -------- | ----- | ----- | ----- | ----- |
> | PSNR↑    | 23.69 | 24.12 | 24.40 | 24.89 |
> | SSIM↑    | 0.653 | 0.592 | 0.617 | 0.722 |
> | LPIPS↓   | 0.304 | 0.351 | 0.349 | 0.238 |
>
> ___
>
> **Q4: Requirements for the input video.**
>
> Thank you for spotting this. We will further include the requirements in the final version of the main paper.
> We use **synchronized light-stage cameras** with ~30–60° angular intervals. While the pipeline works best with moderate viewpoint coverage (≥90° in total), it can tolerate partial occlusions due to the Perspective-Aware Gaussian Transformation. We apply VHAP to perform mesh tracking and extract pose and expression parameters on the datasets. Last, there is no hard constraint on input video length.
>
> ___
>
> **Q5: Discussions on variations in lighting conditions.**
>
> Thank you for pointing this out. Our current work primarily focuses on dynamic reconstruction under controlled indoor lighting and does **not explicitly support lighting variations**. We will further include this discussion in the revised supplementary material.
> Lighting changes can indeed affect 3DGS-based methods due to their reliance on photometric consistency. However, our Jacobian-guided deformation and perspective-aware Gaussian transformation help alleviate blending artifacts caused by view-dependent reflectance, particularly on skin surfaces. These components enhance geometric stability and reduce artifacts under moderate illumination changes.

---

### Official Review · Reviewer_e15x · 2025-07-10

**Clarity:** 3
**Significance:** 3
**Originality:** 4
**Rating:** 5
**Confidence:** 4

**Summary:**

This submission aims to improve geometric accuracy for mesh-rigged Gaussian avatars trained on multiview videos. It tackles the problem that existing works ignore perspective effects when deforming mesh-rigged Gaussians.

The technical contributions are twofold. First, this work introduces perspective-aware projections in 2DGS head avatars. Second, it proposes to use BVH trees to hierarchically organize Gaussian Primitives and quickly extract meshes after training the avatar.

The paper performs quantitative evaluations of geometry on the Multiface dataset and qualitative comparisons on NeRSemble and NHA, comparing with six related works, reporting geometry metrics.

**Questions:**

- What deformations still cannot be modeled with the new formulation and why?
- How does the focal length affect the performance improvements for the proposed homogeneous formulations for the Gaussian projection? Would the effects for smartphones (small focal length) be even greater?
- Why did the authors choose not to report PSNR, SSIM, LPIPS, and not to show more visual results?

**Ethical Concerns:**

["NO or VERY MINOR ethics concerns only"]

**Final Justification:**

This work tackles an important weakness when deforming mesh-rigged Gaussians. The technical contributions (perspective-aware projections and BVH tree for fast mesh extraction) are solid and demonstrate good results.

**Limitations:**

There is no discussion of limitations. I would encourage the authors to openly communicate the failure cases and show examples.

There is no discussion of the societal impact. This could be included in the supp. mat.

**Paper Formatting Concerns:**

n.a.

**Quality:**

3

**Strengths And Weaknesses:**

Strengths

- This work tackles an important weakness when deforming mesh-rigged Gaussians. Previous approaches only consider affine transformations, but perspective transformations are the more correct way of modeling these deformations.
- Obtaining more accurate geometries and adding facial details to expressions is desirable. Given the popularity of Gaussian-splatting-based avatars, this work could have a substantial impact on the research community.
- The mesh extraction is very fast thanks to identifying hopping points using the BVH tree.
- The paper performs quantitative comparisons with six related works and outperforms them in geometry metrics.
- The ablation table clearly shows the improvements when introducing perspective-aware transformations (Tbl. 2).
- Fig. 2 and 3 are helpful to understand the pipeline and impact of the affine vs. perspective transformations.


Weaknesses
- The submission shows surprisingly few qualitative results. There is only one qualitative comparison on meshes in Fig. 5 and a tiny comparison on RGB renderings in Fig. 6. Why didn't the authors add more visuals in the supp. mat? It would be very important to include more examples, specifically video results, in the supp. mat. I observe that the formulation seems to struggle with hair. In the bottom row in Fig. 5, the reconstructed mesh ("Ours") fails to reconstruct the hair and models it as skin surface instead. Showing more qualitative results would help a reader better understand the benefits and limitations of the proposed method.
- The quantitative comparisons do not include reconstruction metrics like PSNR, SSIM, and LPIPS. I would recommend including them, since this is also done in related works [4].
- It would be important to discuss strengths and limitations in more detail. What cannot be modeled with the new formulation and why? It would also be interesting to ablate the impact of the focal length and quantify the benefits of the new formulation. How does the focal length affect the performance improvements for the proposed homogeneous formulations for the Gaussian projection? Would the effects for smartphones (small focal length) be even greater? This could be evaluated on synthetic data or rendered scans from Multiface.

Minor Weaknesses / Suggestions to Authors
- Clarity in Eq. 3: The variable J should be introduced. I had to look into [21] to understand its meaning. Similarly, E should be described in more detail and ideally spelled out explicitly.
- Fig. 5 has the zoom-ins cut off for "Ours".
- The capitalization is not always consistent, e.g., "perspective-aware" (l. 33) vs. "Perspective-aware" (l. 35).
- l. 119 should cite the works that rely on affine approximations.



Conclusion

The paper tackles and important problem and shows promising quantitative results. However, the submission should provide many more qualitative examples and discuss limitations to help a reader understand the impact of the proposed formulation and its potential downsides.

---

> ### Author Rebuttal · Authors · 2025-07-30
>
> Thank you for your valuable feedback and for recognizing our technical contributions toward improving geometric fidelity and extraction speed. We have carefully considered your suggestions and will revise the final version of the paper and supplementary materials accordingly.
>
> ___
>
> **Q1: Additional qualitative results.**
>
> We apologize for the limited number of qualitative examples shown in the main paper and for not including videos in the supplementary material. Due to NeurIPS rebuttal guidelines, we are unable to upload additional images or video results at this stage. However, we have already prepared extensive visual results on the **recently released NeRSemble V2 dataset** and will include them in the revised version along with the supplementary material.
>
> To quantitatively demonstrate the effectiveness of our proposed method in terms of reconstruction quality, we have included additional comparative experiments. Specifically, we conduct several evaluations against state-of-the-art methods, GaussianAvatars (GA) and SurFhead (SF), on tasks including novel-view synthesis, self-reenactment, and temporal consistency. We report overall average metrics and further showcase per-subject qualitative fidelity using randomly selected Subjects #442, #487, and #537.
>
> - Novel-view synthesis.
>
>   We conduct novel-view synthesis experiments and achieve the best results among all compared methods.
>
>   | Methods  | GA    | SF    | Ours      | GA    | SF    | Ours      | GA    | SF    | Ours      | GA     | SF     | Ours      |
>   | -------- | ----- | ----- | --------- | ----- | ----- | --------- | ----- | ----- | --------- | ------ | ------ | --------- |
>   | Subjects | #442  | #442  | #442      | #487  | #487  | #487      | #537  | #537  | #537      | Avg.V2 | Avg.V2 | Avg.V2    |
>   | PSNR↑    | 29.85 | 29.42 | **31.42** | 29.04 | 27.65 | **30.76** | 31.52 | 29.03 | **31.97** | 30.43  | 29.97  | **31.37** |
>   | SSIM↑    | 0.948 | 0.942 | **0.962** | 0.933 | 0.923 | **0.942** | 0.938 | 0.923 | **0.953** | 0.934  | 0.931  | **0.952** |
>   | LPIPS↓   | 0.121 | 0.125 | **0.054** | 0.072 | 0.073 | **0.063** | 0.057 | 0.131 | **0.056** | 0.067  | 0.087  | **0.060** |
>
> - Self-reenactment.
>
>   We conduct self-reenactment experiments, and our method consistently outperforms prior methods in terms of PSNR, SSIM, and LPIPS.
>
>   | Methods  | GA    | SF    | Ours      | GA    | SF    | Ours      | GA    | SF    | Ours      | GA     | SF     | Ours      |
>   | -------- | ----- | ----- | --------- | ----- | ----- | --------- | ----- | ----- | --------- | ------ | ------ | --------- |
>   | Subjects | #442  | #442  | #442      | #487  | #487  | #487      | #537  | #537  | #537      | Avg.V2 | Avg.V2 | Avg.V2    |
>   | PSNR↑    | 25.47 | 25.27 | **25.61** | 26.65 | 26.97 | **27.06** | 25.12 | 25.75 | **26.48** | 25.98  | 25.97  | **26.42** |
>   | SSIM↑    | 0.904 | 0.898 | **0.916** | 0.912 | 0.919 | **0.922** | 0.851 | 0.887 | **0.902** | 0.901  | 0.903  | **0.912** |
>   | LPIPS↓   | 0.147 | 0.149 | **0.098** | 0.081 | 0.077 | **0.074** | 0.101 | 0.139 | **0.077** | 0.081  | 0.098  | **0.073** |
>
> - Temporal consistency.
>
>   To further assess the visual quality of our proposed method, we additionally evaluate our results on **temporal consistency using VMAF[1]**,  a metric designed to capture both perceptual quality and temporal coherence. Our method achieves the highest VMAF scores across all sequences.
>
>   |          | #442-GA | #442-SF | #442-Ours | #487-GA | #487-SF | #487-Ours | #537-GA | #537-SF | #537-Ours | Avg.-GA | Avg.-SF | Avg.-Ours |
>   | -------- | ------- | ------- | --------- | ------- | ------- | --------- | ------- | ------- | --------- | ------- | ------- | --------- |
>   | VMAF[1]↑ | 51.3    | 45.9    | **67.4**  | 49.5    | 44.3    | **66.2**  | 51.0    | 41.6    | **67.9**  | 50.8    | 42.7    | **67.2**  |
>
> Additionally, regarding your concern about the "hair and skin error" in the bottom row of our result in Fig. 5, we believe this may be a misunderstanding caused by the limited viewpoint shown in the "Input RGB," which only displays a frontal view. The subject in question is Subject #253 from the NeRSemble dataset. From other viewpoints, we can tell that our method correctly reconstructs the hair and does not misinterpret it as skin. Moreover, the average calculation is based on multiple frames of the reconstructed results, which can also verify the accuracy of the reconstruction.
>
> We hope that the additional quantitative experiments and clarifications we have provided help to resolve your concern. As you rightly pointed out, this misunderstanding likely stems from the lack of sufficient qualitative comparisons. To address this, we will include more comprehensive visual results and supplementary material to further support our claims.
>
> ___
>
> **Q2: No evaluation on photometric metrics (PSNR / SSIM / LPIPS).**
>
> **3DGS-rendering**: We report the photometric metrics PSNR and SSIM in Tab. 1 of the original manuscript. We additionally compute LPIPS and obtain **0.057** on the NeRSemble dataset (V1), which achieves the best compared to SOTA methods.
>
> **Mesh-based rendering**: We evaluate on four subjects (from Fig. 5) from the frontal view. Currently, TGA does not consider intrinsic decomposition or reflectance modeling. Therefore, we adopt **flat shading based solely on vertex color for rendering**.)
>
> | Subjects | #104  | #264  | #302  | #140  |
> | -------- | ----- | ----- | ----- | ----- |
> | PSNR↑    | 23.69 | 24.12 | 24.40 | 24.89 |
> | SSIM↑    | 0.653 | 0.592 | 0.617 | 0.722 |
> | LPIPS↓   | 0.304 | 0.351 | 0.349 | 0.238 |
>
>
> ___
>
> **Q3: What cannot be modeled with the new formulation?**
>
> We appreciate your suggestion to discuss the limitations of our formulation. The proposed perspective-aware Gaussian transformation faces challenges in specific regions, such as hair and eyes.
> Specifically, **hair** regions are difficult to model accurately due to their translucency and non-rigid nature, which violates the assumptions of FLAME-based mesh tracking. As a result, Gaussian primitives in these areas may exhibit unstable coverage or opacity inconsistency under view changes.
> Eye regions, particularly the **eyeballs**, are also problematic. Our method may fail to produce faithful approximations of the spherical geometry of the eyeball, often resulting in hollow-eye. This is primarily due to the high specularity of the multi-layered cornea, which introduces sharp photometric highlights. These are difficult to fit accurately using the current Spherical Harmonics (SH)-based appearance model. We will include the discussion of these limitations in the revised version of the paper.
>
> ___
>
> **Q4: Impact of the focal length.**
>
> Thank you for this insightful question. Shorter focal lengths (i.e., wider fields of view) introduce stronger perspective distortion, making the affine approximation less accurate. Our homogeneous formulation addresses this by incorporating full-depth information, enabling **more accurate color blending** and better geometry reconstruction under wide-FoV conditions. We will conduct experiments on rendered scans from the on FaceTalk [2] synthetic dataset's meshes and textures with small focal length and include them in the final version of the supplementary material.
>
> ___
>
> **Q5: Discussion of failure cases.**
>
> We will include this discussion and visual examples in the supplemental material.
>
> One failure case occurs under extremely **rapid expressions movements**, such as sudden tongue protrusion or exaggerated frowning. In such cases, local mesh tearing or topological distortions may appear.  Our BVH-based hopping Gaussians strategy assumes relatively smooth and continuous deformations across frames, and is therefore less effective when the motion involves abrupt or large-scale topological changes.
>
> Additionally, such rapid expression changes may temporarily degrade the quality of incremental triangulation due to inaccurate detection of active regions. This can be solved by re-triangulating points after the motion, which helps restore mesh consistency and correct any topology artifacts.
>
> ___
>
> **Q6: Discussion of the societal impact.**
>
> We will include this discussion in the supplemental material.
>
> Our method advances high-fidelity facial reconstruction, but it also poses potential risks of misuse, such as identity theft, unauthorized avatar replication, or deepfake generation. These concerns call for thoughtful reflection on ethical implications and the adoption of practical safeguards to minimize possible harm. But with proper and responsible use, we believe our method can offer significant benefits across various applications, including virtual reality, augmented reality, and the entertainment industry.
>
>
> ___
>
> **Q7: Variable definitions and minor issues (figure cropping, capitalization, citations).**
>
> Thank you for pointing this out. We will correct them in the final manuscript.
>
> ---
>
> **Reference**
>
> [1] Toward a practical perceptual video quality metric.
>
> [2] FaceTalk: Audio-Driven Motion Diffusion for Neural Parametric Head Models.

---

### Note · Authors · 2025-08-14

We sincerely thank all the reviewers for their thoughtful feedback. We are encouraged that our work TGA was recognized for advancing dynamic avatar mesh reconstruction, and we appreciate the positive remarks on:

- Introducing a Perspective-aware Gaussian Transformation that unifies Jacobian-guided deformation with a homogeneous projection formulation, improving projection accuracy, preserving geometric fidelity, and enhancing geometry–color alignment under dynamic conditions.
- Designing an efficient dynamic BVH incremental triangulation strategy that leverages rotation-based updates to detect geometry-changing regions, enabling fast mesh extraction while avoiding unnecessary re-triangulation.
- Achieving strong and consistent results across multiple benchmarks, with clear improvements in geometric accuracy, fine-detail preservation, and inference efficiency compared to state-of-the-art dynamic avatar reconstruction methods.

During the rebuttal phase, we carefully addressed reviewers' concerns and expanded both qualitative and quantitative evaluations to strengthen the completeness of our study. These results will be included in the final supplementary. Specifically:

- Visual Results: We have conducted additional experiments on the **newly released NeRSemble V2 dataset**, providing more visualizations for both novel-view synthesis and self-reenactment. Our method better handles fine structures compared to state-of-the-art dynamic avatar reconstruction baselines.
- Expanded Baseline Comparisons: We have implemented enhanced EWA splatting **variants methods modified with FLAME-Gaussian deformation** for a fair comparison under dynamic facial reconstruction setting. Our approach shows lower geometric and photometric error compared to vanilla 3DGS and these improved baselines.
- Limitations and Impact: We will include a dedicated discussion of the method’s limitations and potential societal implications in the final version of the paper.

Overall, our proposed method effectively addresses the challenge of achieving geometrically accurate and robust dynamic facial reconstruction, a problem that has long hindered both practical applications and methodological advances in the field. We believe these contributions hold substantial value for advancing research and practice in 4D Gaussian-based avatar reconstruction and related domains. We sincerely thank all reviewers for their time, constructive feedback, and recognition of our work.

---

### Decision · Program_Chairs · 2025-09-17

**Decision:**

Accept (spotlight)

**Comment:**

This paper introduces TGA, a framework for dynamic avatar mesh reconstruction that combines a perspective-aware Gaussian transformation with an efficient BVH-based incremental triangulation strategy. The approach directly addresses limitations of prior 3D Gaussian Splatting methods by improving projection accuracy, preserving fine geometric details, and enabling fast and consistent mesh extraction. Reviewers found the technical contributions sound and well-motivated, with quantitative results across multiple benchmarks showing clear improvements in geometric fidelity, temporal consistency, and inference efficiency.

The rebuttal and final remarks were thorough and convincing. The authors went beyond clarifying specific questions by expanding both qualitative and quantitative evaluations, including experiments on the NeRSemble V2 dataset and fairer comparisons with enhanced EWA-splatting baselines. They also discussed limitations in detail. These additions consistently demonstrated superior geometric accuracy, fine-detail preservation, and computational efficiency, which reinforce the significance and robustness of the proposed approach.

While one reviewer remained cautious, calling for broader comparisons and more qualitative visualization, the overall consensus is that this work is novel, timely, and impactful. I strongly recommend acceptance.